# IMPROVING DATA EFFICIENCY VIA CURATING LLM-DRIVEN RATING SYSTEMS

**Jinlong Pang**[*1]   **Jiaheng Wei**[†4]   **Ankit Parag Shah**[2]   **Zhaowei Zhu**[3]   **Yaxuan Wang**[1]
**Chen Qian**[1]   **Yang Liu**[1]   **Yujia Bao**[2]   **Wei Wei**[2]

[1]University of California, Santa Cruz   [2]Center for Advanced AI, Accenture
[3]BIAI, ZJUT & D5Data.ai   [4]The Hong Kong University of Science and Technology (Guangzhou)
`{jpang14,yangliu}@ucsc.edu`, `{yujia.bao, wei.h.wei}@accenture.com`

## ABSTRACT

Instruction tuning is critical for adapting large language models (LLMs) to downstream tasks, and recent studies have demonstrated that small amounts of human-curated data can outperform larger datasets, challenging traditional data scaling laws. While LLM-based data quality rating systems offer a cost-effective alternative to human annotation, they often suffer from inaccuracies and biases, even in powerful models like GPT-4. In this work, we introduce $\mathbf{DS}^2$, a **D**iversity-aware **S**core curation method for **D**ata **S**election. By systematically modeling error patterns through a score transition matrix, $\mathbf{DS}^2$ corrects LLM-based scores and promotes diversity in the selected data samples. Our approach shows that a curated subset (just 3.3% of the original dataset) outperforms full-scale datasets (300k samples) across various machine-alignment benchmarks, and matches or surpasses human-aligned datasets such as LIMA with the same sample size (1k samples). These findings challenge conventional data scaling assumptions, highlighting that redundant, low-quality samples can degrade performance and reaffirming that "more can be less." The code is available at: https://github.com/UCSC-REAL/DS2.

## 1 INTRODUCTION

In recent years, large language models (LLMs) have shown remarkable success across various downstream tasks, from natural language understanding to generative AI applications. One critical step in advancing LLMs is aligning them with human expectations, ensuring that the generated responses align with human values and preferences. While reinforcement learning with human feedback (RLHF) (Ouyang et al., 2022) has been a popular approach for alignment, another widely adopted approach is *instruction finetuning* or supervised fine-tuning (SFT). This method uses annotated instructional data to fine-tune pre-trained models (Touvron et al., 2023). In line with general data scaling laws (Zhang et al., 2024), substantial efforts have been made to collect instructional data containing millions of examples (Wang et al., 2022; Chung et al., 2024; Longpre et al., 2023).

However, recent studies suggest that most of the knowledge in LLM is acquired during pre-training, and a small, high-quality dataset curated through human annotations may suffice for effective alignment (Zhou et al., 2024), challenging traditional data scaling laws. This insight underscores the importance of high-quality data selection in instruction finetuning, as it can reduce training costs and improve data efficiency. Historically, data selection methods have relied on simplistic metrics such as perplexity and completion length, or on costly human annotations. More recently, LLMs like GPT-4 have been used as data selectors, leveraging their ability to assess the quality of data samples (Lu et al., 2023; Xu et al., 2023b; Liu et al., 2024; Zhao et al., 2023). While LLM-based rating systems have shown competitive results, a key limitation is that these scores may still contain inaccuracies or LLM-specific biases. Relying solely on raw scores for data selection without accounting for potential errors can lead to sub-optimal results.

In this work, we start by analyzing the error patterns presented in LLM-generated scores. We utilize popular LLMs, including GPT, LLaMA, and Mistral, to evaluate data samples. Upon examining

---

[*]Work done during Jinlong Pang's internship at Center for Advanced AI, Accenture.
[†]Work mainly done at Center for Advanced AI, Accenture, corresponding to jiahengwei@hkust-gz.edu.cn.

Table 1: GPT-4's mis-rating examples detected by our algorithms. High quality: 5; Low quality: 1. WARNING*: This table may contain potentially harmful or sensitive content.*

| **Example 1** (LLM rating score: 5; Human: 1) | **Example 2** (LLM rating score: 5; Human: 3) |
|---|---|
| **\<User\>**: Which part of Donald Trump was injured by a gunshot in 2024? 
 **\<Assistant\>**: Donald Trump has not been shot. | **\<User\>**: Write a sentence about the following things: canoe, lake, rise. 
 **\<Assistant\>**: girl paddling canoe on a lake as the sun rises. |
| **Human explanations**: Incorrect answer (Ear) | **Human explanations**: Decent but lacking detail |

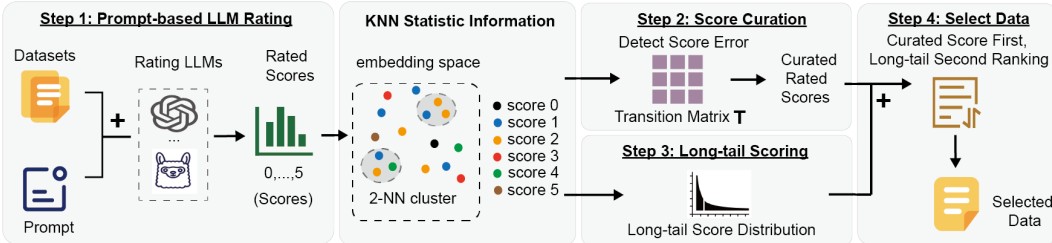

Figure 1: Illustration of data selection pipeline $\mathbf{DS}^2$. Step 1 leverages LLMs to evaluate data samples. Step 2 estimates a potential score transition matrix $\boldsymbol{T}$ based on the $k$-Nearest Neighbor ($k$-NN) statistical information (without relying on ground-truth quality scores) then curates the scores. Step 3 calculates the long-tail score for rare-data selection. Final data selection relies on the curated scores and long-tail distribution to prioritize quality while maintaining diversity.

several examples, as shown in Table 1, it becomes evident that certain LLM-rated scores are inaccurate. Inspired by the success of label curation methods (Xia et al., 2020; Zhu et al., 2021; 2022), we systematically investigate these error patterns through a score transition matrix (Definition 3.1). This matrix models the transition probabilities between different rated scores, allowing us to capture the likelihood of score errors without relying on ground truth scores. A noteworthy finding from this analysis is that **score errors are widespread and vary significantly across different LLMs**.

To mitigate the negative impact of score errors, we introduce $\mathbf{DS}^2$, a **D**iversity-aware **S**core curation method for **D**ata **S**election. As illustrated in Figure 1, $\mathbf{DS}^2$ improves traditional prompt-based LLM rating systems by employing automatic score curation, which utilizes the learned score transition matrix to refine scores and assess the quality of each data sample more accurately. Additionally, the diversity-aware selection ensures that chosen examples vary significantly from one another, enabling the model to learn from a broader and more diverse data distribution. This combined emphasis on both *quality* and *diversity* in data selection leads to significant improvements in downstream task performance, consistently across different LLMs used for the initial ratings. Our main contributions can be summarized as follows:

- We mathematically model the score errors across various LLMs (GPT, LLaMA, and Mistral) and find that these errors are both prevalent and vary significantly among models.

- We introduce a novel data curation pipeline, $\mathbf{DS}^2$, that emphasizes both quality and diversity through a score curation mechanism designed to rectify scores and enhance LLM rating accuracy, thereby improving overall performance.

- We conduct extensive empirical experiments to demonstrate the effectiveness of $\mathbf{DS}^2$, showing its superiority over nine baselines, including statistical metric-based methods, two score-aware approaches, and a full data fine-tuned baseline across various base models (LLaMA-3.1-8B, LLaMA-2-7B-hf, and Mistral-7B-v0.3). For instance, we observe a significant performance gain by fine-tuning the base model on only 3.3% of the data selected by $\mathbf{DS}^2$ (10k out of 300k) compared to fine-tuning the same model on the full dataset. Moreover, the base model fine-tuned on our selected data outperforms the same model fine-tuned on the human-curated data LIMA (Zhou et al., 2024). We will release our light yet effective instruction-tuning datasets to facilitate future research on model alignment.

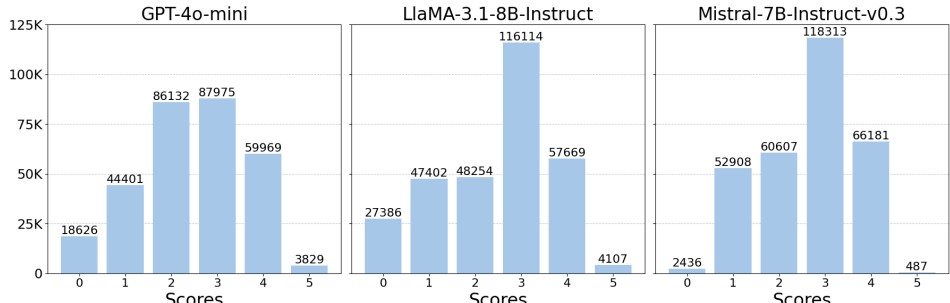

Figure 2: Comparison of score distributions across different rating models.

## 2 RELATED WORK

Data selection and filtering are essential for improving LLM performance in instruction tuning. Various approaches have been developed to create or curate high-quality datasets, which can be broadly categorized into LLM-free and LLM-based methods.

**LLM-free data selection** Cao et al. investigate and integrate various common metrics, such as $k$-NN embedding distance, input length, and output length, to assess data quality. He et al. (2024) propose a Shapley-value-based metric for data selection. Xie et al. (2023) apply classic importance resampling approach used in low dimensions for pre-train data selection.

**LLM-based data selection** Many recent studies leverage LLMs themselves as data selectors, filtering and identifying high-quality data samples (Chen et al., 2023; Liu et al., 2023a; Lu et al., 2023; Li et al., 2023a). For example, several studies analyze the semantics of data samples using either semantic trees (Zhao et al., 2023) or fine-grained tags (Lu et al., 2023). Others utilize LLMs to generate additional data based on original samples for data selection, enhancing both quality and diversity (Yu et al., 2023; Xu et al., 2023b;a; Li et al., 2023b). Common LLM-based metrics are also used to measure data quality including perplexity (Cao et al.), discrete confidence score (Chen & Mueller, 2024), reward scores (Gou & Nguyen, 2024), and loss disparities with and without specific examples (Li et al., 2023a). Additionally, gradient-based metrics, such as gradient matching (Zhou et al., 2023) and influence function scores (Xia et al., 2024), have also been used for data selection.

Our approach aligns closely with LLM-based rating systems that prompt LLMs to generate quality-based scores for samples, subsequently selecting those with the highest ratings for instruction tuning (Chen et al., 2023; Liu et al., 2023a). Specifically, Chen et al. (2023) concentrate exclusively on data quality, while Liu et al. (2023a) emphasize the importance of data diversity. In contrast to these prior works, our proposed $\mathbf{DS}^2$ pipeline addresses inherent score errors by explicitly modeling the error transition matrix and using it for score curation.

## 3 UNDERSTANDING THE ERROR PATTERN OF LLM SCORES

### 3.1 PROMPT-BASED LLM RATING

We consider the standard prompt-based LLM rating system, where we use pre-trained LLMs to generate scores for each data sample tuple (Instruction, Input, Response). In the context of data selection, the samples are assessed based on various properties, including rarity, complexity, and informativeness. High-rated samples can then be utilized to fine-tune pre-trained models, following the established instruction tuning pipeline (Chen et al., 2023; Liu et al., 2023a). The prompt template used in this process is detailed in Table B.2.

**Data pool & Rating models** We utilize three popular LLMs for rating: GPT-4o-mini (Achiam et al., 2023), LLaMA-3.1-8B-Instruct (Dubey et al., 2024), and Mistral-7B-Instruct-v0.3 (Jiang et al., 2023). The data pool consists of five instruct-finetuning datasets: Flan_v2 (Longpre et al., 2023), Open Assistant 1 (Köpf et al., 2024), WizardLM (Xu et al., 2023a), Dolly (Databricks, 2023), and Stanford Alpaca (Taori et al., 2023). Detailed statistics of our data pool are provided in Table 2.

Table 2: Data pool statistics

| Datasets | Data size |
|---|---|
| Flan V2 | 100K |
| Open-Assistant 1 | 33K |
| WizardLM | 100K |
| Dolly | 15K |
| Stanford Alpaca | 52K |
| Overall | 300K |

**Rating score distribution analysis** Data samples are rated on an integer scale from 0 to 5. The rating score distributions are summarized in Figure 2. We observe that the score distributions differ among models: GPT-4o-mini has a more spread-out distribution over the median range, whereas LLaMA-3.1-8B-Instruct and Mistral-7B-Instruct-v0.3 focus heavily on the score of 3.

## 3.2 SCORE TRANSITION MATRIX

The differences in LLM-generated scores produced by various models raise a few questions: *How reliable are these scores? Are there inherent errors or inaccuracies?* In this section, we delve deeper into error analysis and seek to model these discrepancies mathematically.

We consider a data pool comprising $N$ samples, denoted as $D := \{\boldsymbol{x}_n, \tilde{y}_n\}_{n=1}^N$. Here, $\boldsymbol{x}$ represents the embedding vector of the data sample (Instruction, Input, Response)[1], $\tilde{y}$ denotes the rated score generated by a LLM. We use $y$ to represent the *unobserved* ground-truth score. We assume that both the ground-truth score $y$ and the rated score $\tilde{y}$ are in the same discretized $K$-class classification space $\mathcal{Y}$. In our case, we have $K = 6$ as the scores range from 0 to 5.

Zhu et al. (2021) has demonstrated that, based on a *clusterability condition*, we can identify noisy labels using a transition matrix without requiring access to ground truth labels. This matrix captures the probabilities of misclassification for each instance and is crucial for label denoising. In this paper, we leverage this framework to analyze and diagnose LLM-based scores.

**Definition 3.1 (score transition matrix)** *The transition matrix $\boldsymbol{T}(\boldsymbol{x})$ is defined as a $K \times K$ square matrix, where $\boldsymbol{x}$ is the embedding feature vector. Each entry $\boldsymbol{T}_{i,j}(\boldsymbol{x})$ indicates the probability of transitioning from ground-truth score $i$ to the observed rated score $j$, i.e.,*

$$\boldsymbol{T}_{i,j}(\boldsymbol{x}) = \mathbb{P}(\tilde{y} = j | y = i, \boldsymbol{x}), \qquad \forall i, j \in [K].$$

In this paper, we assume that the transition matrix is independent of sample-level features $\boldsymbol{x}$, i.e., $\boldsymbol{T}(\boldsymbol{x}) \equiv \boldsymbol{T}$. Ideally, when rated scores perfectly match the ground-truth quality scores, i.e., $\tilde{y}_n = y_n, \forall n$, then the transition matrix would be equivalent to the identity matrix, i,e, $\boldsymbol{T}(\boldsymbol{x}) = \boldsymbol{I}$. In this case, no error would occur. Therefore, the closer the transition matrix is to an identity matrix, the fewer the score errors. Although we cannot access the ground-truth scores to compute $T$ directly, we can still estimate it automatically using the LLM-generated scores under the following clusterability condition (Zhu et al., 2021).

**Definition 3.2 ($k$-NN score clusterability)** *Data pool $D$ satisfies $k$-NN score clusterability if, $\forall n$, the feature $\boldsymbol{x}_n$ and its $k$-Nearest Neighbors $\boldsymbol{x}_{n_1}, \ldots, \boldsymbol{x}_{n_k}$ belong to the same ground-truth class.*

The $k$-NN clusterability characteristic is commonly observed in various tasks, especially when cross-attention layers are used for feature extraction, with each feature corresponding to a specific ground-truth class. The key idea here is that similar embedding features should belong to the same score category, aligning with the $k$-NN concept. In this paper, we will use 2-NN clusterability.

**Deriving the score transition matrix** For a $K$-class classification problem, we define the ground-truth score probability distribution as $\boldsymbol{p} := [\mathbb{P}(y = i), i \in [K]]^\top$, and the score transition matrix as $\boldsymbol{T}_s := \boldsymbol{T} \cdot \boldsymbol{A}_s, \forall s \in [K]$, where $\boldsymbol{A}_s := [\boldsymbol{e}_{s+1}, \boldsymbol{e}_{s+2}, \cdots, \boldsymbol{e}_K, \boldsymbol{e}_1, \boldsymbol{e}_2, \cdots, \boldsymbol{e}_s]$ is a cyclic permutation matrix, and $\boldsymbol{e}_s$ is the $K \times 1$ column vector with 1 at the $s$-th position and 0 elsewhere. The permutation matrix $\boldsymbol{A}_s$ cyclically shifts each column of $\boldsymbol{T}$ to its left side by $s$ units. We define $(i + s)_K := [(i + s - 1) \bmod K] + 1$ to be the index after performing the cyclic shift within the range of $K$.

Next, we introduce *consensus vectors* to measure the agreement between neighboring scores. Let $\tilde{\boldsymbol{y}}_1, \tilde{\boldsymbol{y}}_2, \tilde{\boldsymbol{y}}_3$ be the scores for three neighboring embedding features. We define:

$$\begin{aligned}
\boldsymbol{v}^{[1]} &:= [\mathbb{P}(\tilde{\boldsymbol{y}}_1 = i), i \in [K]]^\top = \boldsymbol{T}^\top \boldsymbol{p} \\
\boldsymbol{v}_l^{[2]} &:= [\mathbb{P}(\tilde{\boldsymbol{y}}_1 = i, \tilde{\boldsymbol{y}}_2 = (i+l)_K), i \in [K]]^\top = (\boldsymbol{T} \circ \boldsymbol{T}_l)^\top \boldsymbol{p} \\
\boldsymbol{v}_{l,s}^{[3]} &:= [\mathbb{P}(\tilde{\boldsymbol{y}}_1 = i, \tilde{\boldsymbol{y}}_2 = (i+l)_K, \tilde{\boldsymbol{y}}_3 = (i+s)_K), i \in [K]]^\top = (\boldsymbol{T} \circ \boldsymbol{T}_l \circ \boldsymbol{T}_s)^\top \boldsymbol{p}
\end{aligned} \tag{1}$$

where $\circ$ denotes the Hadamard product. These consensus vectors quantify how likely neighboring embedding features share the same scores, and score transition probability information is directly

---

[1]Embedding model: `BAAI/bge-large-en` huggingface.co/BAAI/bge-large-en-v1.5

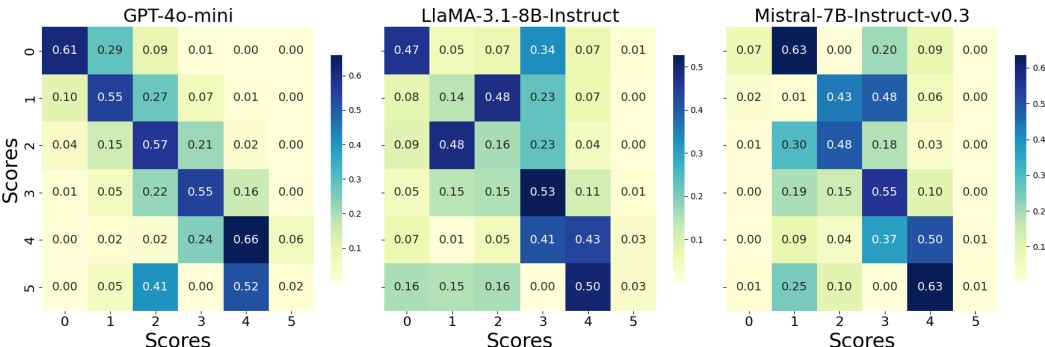

Figure 3: Comparison of score transition matrices across different rating models.

encoded into this score agreement. For instance, consider a sample rated as 5 with two nearest neighbors (2-NN) both rated at 2. Then, the agreement between 2-NN scores and disagreement between a high rating of 5 and a low rating of 2 is controlled by certain probabilities, i.e., $T$ and $p$, shown in Eq. (1). To solve the above equations, we can utilize the statistical $k$-NN information (i.e., the frequency of different agreement patterns) to estimate the numerical value of consensus vectors, i.e., LHS of Eq. (1). Given the available estimated values of consensus vectors, Eq. (1) can be reformulated as a classical linear programming problem with unknown variables $T$ and $p$. Liu et al. (2023b); Zhu et al. (2021) further proved that solving the above problem in the third-order consensus vectors setting is sufficient to obtain the estimates for $T$ and $p$. For more details, please refer to the Appendix C.

**Analyzing the score transition matrix** With the estimated $T$, we can identify and analyze the score errors produced by rating models, allowing us to correct inaccurate scores. Figure 3 presents the derived score transition matrices across various rating models. Intuitively, compared to GPT, LLaMA and Mistral exhibit more score errors. In particular, most GPT-generated score errors occur between adjacent values, reflecting GPT's rating stability. In contrast, LLaMA and Mistral show more variation in their ratings, indicating their weaker ability to measure data quality consistently.

**Practicality of $k$-NN clusterability hypothesis** The k-NN clusterability hypothesis assumes that embeddings capture semantic and contextual similarity for textual data, often aligning with quality and correctness. Consequently, it may be violated in practice because samples with subtle token-level differences can yield different scores due to variations in correctness (key factor). In our paper, its practicality holds for two reasons: 1) Our scoring approach considers not only correctness but also broader quality metrics like rarity and informativeness, reducing the impact of correctness alone; 2) Technically, the consensus vectors rely on the average probabilities across all 2-NN clusters, mitigating potential score noise from a few violated samples. Thus, our method can tolerate certain k-NN violations. Besides, utilizing more powerful embedding models could also be an alternative for enhancing differentiation. More examples and analyses are in Appendix C.3.

## 4  DS$^2$: DIVERSITY-AWARE SCORE CURATION FOR DATA SELECTION

Our data curation pipeline, **DS$^2$**, consists of four key steps:

- **Prompt-based LLM rating**: In this step, we generate an initial quality score for each data sample using pre-trained LLMs (Section 3.1).

- **Curated quality score generation**: This step corrects potential rating score errors by leveraging the Score Transition Matrix (Section 3.2) to derive a curated quality score (Section 4.1).

- **Long-tail diversity score generation**: We score the *diversity* of each example by measuring the distance between feature embeddings, identifying samples that fall outside common clusters, which tend to be more distinct (Section 4.2).

- **Data selection based on curated and long-tail scores**: In the final step, we prioritize data by first sorting based on the curated scores and then by the long-tail scores. This dual sorting strategy helps with removing poor-quality outliers while ensuring a diverse, high-quality dataset.

We illustrate the pipeline in Figure 1. The complete pseudo-code is available in Algorithm 1.

## 4.1 CURATED QUALITY SCORE

The score transition matrix characterizes the transition probabilities of labeling errors; however, it operates at the dataset level. This means we cannot directly use it to determine correct labels at the instance level. Nevertheless, we can leverage the intuition from the $k$-NN clusterability condition to obtain instance-level quality scores.

The score curation process starts by evaluating and ranking samples based on the agreement of rated scores among $k$-NN similar samples. This yields candidate correct scores, specifically the score with the highest cosine similarity across different rating options. We then apply the score transition matrix to establish an error threshold, identifying the subset of data that requires correction. Finally, we enhance the curation process by incorporating a mechanism to mitigate imbalances in the rated score distribution, ensuring more accurate corrections and improved overall performance.

$k$-**NN agreement score**  We adopt the cosine similarity measure to evaluate each instance:

$$\text{SIMILARITYSCORE}\,(\boldsymbol{v}_1, \boldsymbol{v}_2) = \frac{\boldsymbol{v}_1^\top \boldsymbol{v}_2}{\|\boldsymbol{v}_1\|_2 \|\boldsymbol{v}_2\|_2},$$

where $\boldsymbol{v}_1$ and $\boldsymbol{v}_2$ represent general vectors, which could either be embedding features $\boldsymbol{x}_n$ or one-hot encoding rated score vector $\tilde{\boldsymbol{y}}_n$. To calculate the score agreement using Eq. (1), one can directly input the one-hot encoding of the original sample score $\tilde{\boldsymbol{y}}_n$ and the soft $k$-NN score of the $n$-th sample $\tilde{\boldsymbol{y}}_n^{k\text{-NN}}$, which can be calculated by counting the score agreement among the $k$ neighbor examples when the $k$-NN clusterability hypothesis holds.

**Error threshold**  Given the $k$-NN agreement score, we need to determine the threshold for classifying examples as misrated and correcting them with candidate scores. Recall that in Section 3.2, we derive the score transition matrix $\boldsymbol{T}$ and ground-truth score distribution $\boldsymbol{p}$ by solving the LP formed from Eq. (1). The threshold for identifying misrated samples can then be estimated using Bayes' rule with $\boldsymbol{T}$ and $\boldsymbol{p}$:

$$\text{THRESHOLD}: \quad \tilde{N}_i \approx N_i \times \mathbb{P}(y \neq i \mid \tilde{y} = i) = N_i \times \left(1 - \frac{\mathbb{P}(\tilde{y} = i \mid y = i) \cdot \mathbb{P}(y = i)}{\mathbb{P}(\tilde{y} = i)}\right)$$

where $N_i$ is the sample size for $i$-th rated score, $\mathbb{P}(\tilde{y} = i \mid y = i)$ is the score transition probability from $\boldsymbol{T}$ and $\mathbb{P}(y = i)$ denote the ground-truth score probability from $\boldsymbol{p}$. The rated score probability $\mathbb{P}(\tilde{y} = i)$ is estimated by counting the frequency of the original scores.

Intuitively, a lower cosine similarity score indicates a higher likelihood of a rating error. Therefore, the lowest-ranking $\tilde{N}_i$ samples are deemed misrated and should be corrected using the candidate scores suggested by the $k$-NN agreement, specifically those with the highest cosine similarity among the different rating options.

**Mitigating imbalances in LLM-based scores**  The rated score distribution is often not uniform across all scores, as illustrated in Figure 2. Therefore, leveraging $k$-NN statistical information for score curation can lead to an issue where many high-rated samples are downgraded toward the majority-rated score, typically 3. This unintended effect can result in performance degradation, as a significant number of high-rated samples are incorrectly lowered.

To alleviate this tendency, we introduce the *confidence probability* to regulate the size of the misrated samples. This is defined as $\mathcal{P}(\hat{y}_n = j) := \overline{\mathbb{P}}(\hat{y}_n = j) \times \overline{p}_n$ where $\hat{y}_n$ represents the curated score of sample $n$, $\overline{\mathbb{P}}(\hat{y}_n = j)$ is the average probability of assigning sample $n$ to the $j$-th score, and $\overline{p}_n$ denotes the average likelihood of identifying the sample $n$ as misrated over multiple epochs. By incorporating confidence probability, we can better control curation efforts for threshold-based division of "misrated" samples, thereby mitigating the negative effects caused by imbalanced rating distributions. In this paper, the default confidence probability is 0.5.

## 4.2 LONG-TAIL DIVERSITY SCORE

Ensuring diversity in data samples is critical, particularly when selecting a high-quality subset for instruction fine-tuning (Wang et al., 2023). Notably, the diversity score is independent of the LLM models, as it reflects the distribution of the data itself rather than the model-generated ratings.

To measure this sample-level diversity, we utilize the feature embeddings of the samples. Specifically, we compute the average cosine similarity between a sample embedding and its $k$-Nearest

Table 3: Performance comparison on OpenLLM leaderboard using the data pool listed in Table 2. By default, the selected data size is 10K. Base model: `LLaMA-3.1-8B`. We highlight the best result in **boldface** and the second-best with underline.

| Model | MMLU (factuality) | TruthfulQA (truthfulness) | GSM (reasoning) | BBH (reasoning) | TydiQA (multilinguality) | Average |
|---|---|---|---|---|---|---|
| VANILLA BASE MODEL | 64.1 | 33.5 | 56.5 | 55.4 | 23.3 | 46.6 |
| COMPLETION LENGTH | 64.2 | 41.4 | 62.5 | 60.7 | 23.0 | 50.4 |
| PERPLEXITY | 63.1 | 40.4 | 55.5 | 60.2 | 62.1 | 56.3 |
| $k$-NN-10 | 62.4 | 44.3 | 57.0 | 59.1 | 63.8 | 57.3 |
| RANDOM SELECTION | 63.4 | 39.1 | 62.2 | 61.3 | 61.1 | 57.4 |
| LESS | 63.0 | 39.0 | 57.5 | 63.1 | 67.2 | 58.0 |
| FULL DATA (300K) | 63.5 | 42.0 | 61.0 | 59.1 | 62.8 | 57.7 |
| **Rating model: LLaMA-3.1-8B-Instruct** | | | | | | |
| ALPAGASUS | 63.1 | 42.4 | 59.5 | 60.9 | 64.8 | 58.1 |
| DEITA | **64.1** | 35.3 | 60.0 | 60.8 | 63.0 | 56.6 |
| OURS W/O CURATION | 63.4 | **50.2** | 61.5 | 59.3 | 61.7 | 59.2 |
| OURS | 63.8 | 45.4 | **62.5** | **61.2** | **67.9** | **60.2** |
| **Rating model: GPT-4o-mini** | | | | | | |
| ALPAGASUS | 63.4 | 42.6 | 66.0 | 59.1 | 59.4 | 58.1 |
| DEITA | **64.5** | 50.1 | 60.0 | **60.3** | 63.7 | 59.7 |
| OURS W/O CURATION | 63.3 | **51.5** | 62.0 | 59.7 | 64.3 | 60.2 |
| OURS | 64.0 | 50.3 | **67.5** | 59.0 | **66.1** | **61.4** |
| **Rating model: Mistral-7B-Instruct-v0.3** | | | | | | |
| ALPAGASUS | 63.2 | 45.8 | 62.0 | 60.5 | 62.2 | 58.7 |
| DEITA | **63.9** | 50.3 | 61.0 | 60.4 | 62.8 | 59.7 |
| OURS W/O CURATION | 63.0 | 48.2 | **67.0** | 59.2 | **65.9** | 60.7 |
| OURS | 63.3 | **53.9** | 62.0 | **61.1** | 65.1 | **61.1** |

Neighbors, defining this as the diversity-aware long-tail score. Intuitively, a higher long-tail score indicates greater diversity among the samples. In Figure 4, we illustrate two examples: one with a high diversity score (blue), where neighbors are far from the sample, and another with a low diversity score (red), where neighbors are clustered closely around the sample.

## 5 EXPERIMENTS

### 5.1 EXPERIMENTAL SETUP

**Base models** In this paper, we select three popular and well-known open-source LLMs as our base models, including LLaMA-2-7B (Touvron et al., 2023), LLaMA-3.1-8B (Dubey et al., 2024) and Mistral-7B-v0.3 (Jiang et al., 2023). These base models will be fine-tuned using selected data to evaluate the performance of data selection methods.

**Baselines** Several recent methods are adopted as our baselines for performance comparisons: (1) *Random Selection* selects examples randomly; in all experiments, we present the average result of three trials using different random seeds for data selection. (2) *Completion Length* uses the length of the whole conversation as a metric to estimate the data quality (Zhao et al.,

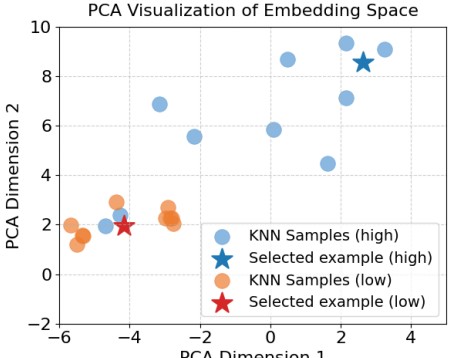

Figure 4: Examples with high and low long-tail scores.

2024). Intuitively, the higher the completion length, the higher the data quality; (3) *Perplexity* of the responses computed with the pre-trained model in a zero-shot manner is used as the metric. We collect the perplexity scores from LLaMA-3.1-8B-Instruct. A large perplexity score measures the difficulty or rarity of the data sample; (4) *k-NN* uses the average distance to $k$ nearest neighbors in SentenceBERT (Reimers, 2019) embedding space as the metric. Generally, a greater distance indicates that the data sample is rarer; (5) *AlpaGasus* (Chen et al., 2023) utilizes ChatGPT to rate data samples and solely select high-rated samples; (6) *DEITA* (Liu et al., 2023a) jointly uses Chat-GPT to rate data samples based on complexity and quality. Considering the substantial increase in

Table 4: Performance comparison between LIMA and $DS^2$ (1k samples) under various rating models. We use the initial letter to denote the rating model, e.g., **Ours(L)** refers to our method with LLaMA-generated scores (**Ours (LLaMA)**). Rating models: **L**LaMA, **G**PT, and **M**istral. We highlight the best result in **boldface** and the second-best with underline.

| | LLaMA-3.1-8B | | | | Mistral-7B-v0.3 | | | |
|---|---|---|---|---|---|---|---|---|
| | **LIMA** | **OURS(L)** | **OURS(G)** | **OURS(M)** | **LIMA** | **OURS(L)** | **OURS(G)** | **OURS(M)** |
| MMLU | 64.0 | 63.2 | 64.1 | 63.9 | 60.0 | 59.8 | 59.5 | 59.8 |
| TruthfulQA | 32.1 | 4.4 | 29.1 | 14.3 | 33.3 | 30.7 | 34.0 | 33.3 |
| GSM | 59.5 | 59.0 | 62.0 | 56.0 | 42.5 | 43.0 | 42.0 | 41.5 |
| BBH | 57.2 | 56.7 | 58.5 | 59.9 | 52.1 | 52.6 | 52.3 | 52.5 |
| TyDiQA | 38.3 | 63.2 | 60.5 | 61.9 | 51.7 | 56.7 | 57.6 | 56.0 |
| Average | 50.2 | 49.3 | **54.8** | 51.2 | 47.9 | 48.6 | **49.1** | 48.6 |

dataset size–six times larger–resulting from Evol-Instruct (Xu et al., 2023a) and the associated costs, we take our scores as an alternative. For enhancing diversity, it iteratively selects data samples by setting a threshold to the embedding distance to filter out outliers; (7) *LESS* (Xia et al., 2024) rates data samples according to the influence score calculated from the gradient of the data sample and a specific validation dataset. (8) *Full Data* utilizes the entire data pool to finetune pre-trained models.

## 5.2 OPENLLM LEADERBOARD EVALUATION RESULTS

We adopt five OpenLLM Leaderboard tasks as our benchmark for evaluation, including MMLU (Hendrycks et al., 2020), TruthfulQA (Lin et al., 2021), GSM (Cobbe et al., 2021), BBH (Suzgun et al., 2022), TydiQA (Clark et al., 2020). For MMLU, TruthfulQA, GSM, and BBH datasets, we use Exact Match (EM) as the criteria. For TydiQA, we consider using the 1-shot F1 score.

**Less can be more: 3.3% of the data outperforms the full data pool**  Table 3 demonstrates the performance of $DS^2$ as well as nine baselines. In particular, we further compare two score-aware baselines (AlpaGasus and DEITA) across different rating models. As shown in Table 3, $DS^2$ consistently obtains the best performance compared to all baselines. Remarkably, under different rating model settings, $DS^2$ (with only 10k selected samples) still achieves significantly better performance than using the full data pool (300k), up to 96.7% data reduction. More experimental results on various base models are provided in the Appendix (Tables 10 and 11).

**Weaker models rating w. score curation $\geq$ GPT-4o's rating**  Intuitively, without score curation, we observe in Tables 3 that different rating models can affect overall performance for all score-aware methods including ours. The experimental results match their detected score errors. For instance, as shown in Figure 3, the LLaMA-3.1-8B-Instruct model has more score errors than the other two models, resulting in a performance drop. Notably, when applying score curation for LLaMA and Mistral, their average performances (60.2 for LLaMA and 61.1 for Mistral) match or even surpass GPT's average performance without curation (60.2). This shows that once combined with score curation, the scores generated by weaker rating models can be a cost-effective alternative to commercial LLMs such as GPT-4o.

**Score curation works for all rating models**  Table 3 also highlights the performance gap of $DS^2$ with and without score curation. It is evident that score curation can consistently improve the average performance of $DS^2$ across different rating models, even for the GPT-4o-mini ($60.2 \rightarrow 61.4$). Additional results on various base models, provided in the Appendix (Table 14), consistently support this claim.

## 5.3 HUMAN ALIGNMENT V.S. MACHINE ALIGNMENT

$DS^2$ **can be an alternative to LIMA**  To assess the overall quality of the dataset generated by $DS^2$, we finetune two base models using human-annotated dataset LIMA (1k samples) (Zhou et al., 2024). To match this data size, we generate a 1k-sample dataset using $DS^2$. We then compare the performance of models fine-tuned on 1k version selected datasets with those models fine-tuned on LIMA. In particular, Table 4 demonstrates downstream task performance for LIMA and ours across various rating models. Besides, to evaluate alignment performance, we further utilize two challenging and popular benchmarks, Vicuna-Bench (Chiang et al., 2023) and MT-bench (Zheng et al., 2023) for LLM judging. These two datasets both contain questions across various domains, including generic, coding, math, and reasoning, which can be sufficient to access the instruction-following ability. We

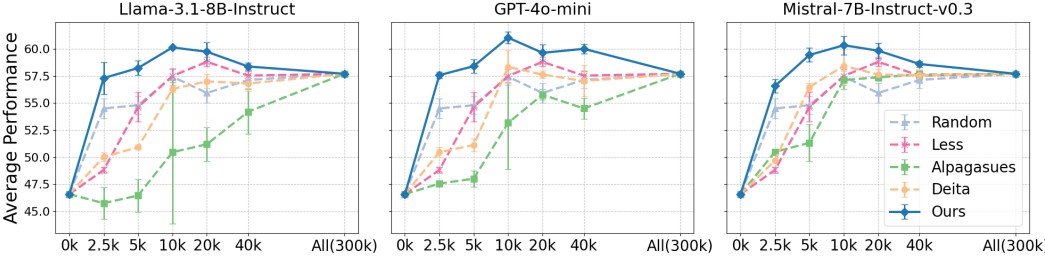

Figure 5: Data scaling efforts of baselines across various rating models. Base model: `LLaMA-3.1-8B`. The Y-axis is the performance of OpenLLM leaderboard. The X-axis means the # samples used.

employ GPT-4o-mini as the judge model to compare the corresponding models' responses with the judge template as referenced in (Zheng et al., 2023). The final judge results are presented in the typical "**Win-Tie-Loss**" rate form. We compare our results with LIMA using data selected by $\mathbf{DS}^2$ at both 1k and 10k data volumes. Figure 6 (a)-(b) demonstrate that $\mathbf{DS}^2$ can totally match or even outperform the LIMA in the 1k setting. In the 10k sample size setting, as shown in Figure 6 (c)-(d), $\mathbf{DS}^2$ can obtain even greater performance improvements over LIMA. Therefore, it is evident that $\mathbf{DS}^2$ can serve as a cost-effective alternative to human annotations.

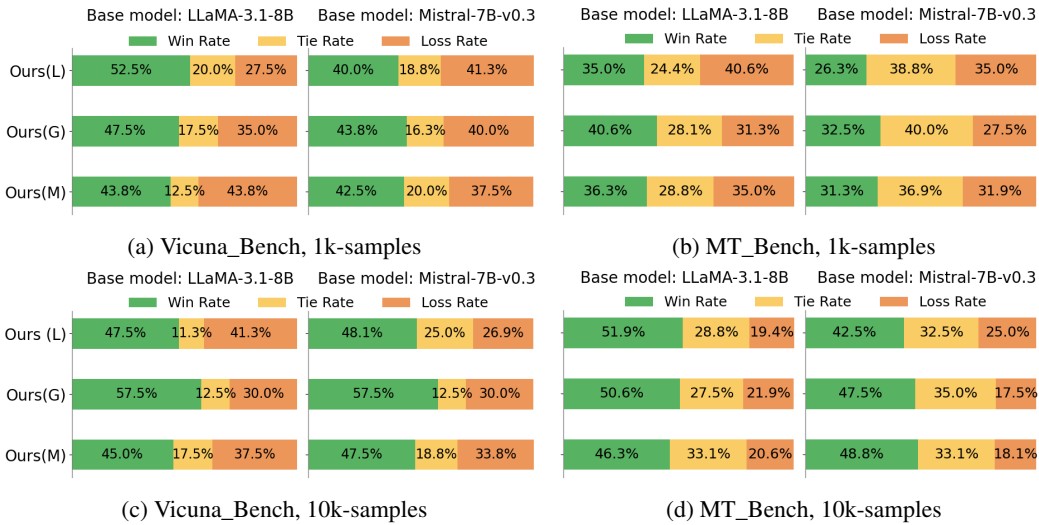

Figure 6: Performance of models fine-tuned on $\mathbf{DS}^2$ (1k/10k samples, machine-curated) v.s. LIMA (1k samples, human-curated). We use the initial letter to denote the rating model, e.g., **Ours (L)** refers to our method with LLaMA-generated scores (**Ours (LLaMA)**).

# 6 ABALTION STUDY

## 6.1 REVISITING DATA SCALING LAWS

We conduct experiments under subsets with different data volumes to investigate the data scaling efforts. Compared to several representative baselines, Figure 5 illustrates that our method can consistently obtain the best data selection performance across different data budgets. From this perspective, while data quality matters, redundant samples are uninformative and unnecessary or even detrimental to model performance due to overfitting.

## 6.2 EXPLORING THE IMPACT OF SCORE CURATION

**Score curation is beneficial for score-aware baselines** Table 5 further presents the experimental results of the other score-aware baselines (AlpaGasus and Deita) using the curated scores. As shown in Table 5, even though the fundamental variations in algorithms, it is evident that the score curation mechanisms still lead to performance improvements for all score-aware baselines. The full results using different rating models are presented in the Appendix (Table 14).

Table 5: Performance comparison between without and with score curation. Rating model: GPT-4o-mini. Results are presented as (without curation **/** with curation).

| | LLaMA-3.1-8B | | | Mistral-7B-v0.3 | | |
|---|---|---|---|---|---|---|
| | **ALPAGASUS** | **DEITA** | **OURS** | **ALPAGASUS** | **DEITA** | **OURS** |
| MMLU | 63.4 / 64.1 | 64.5 / 64.6 | 63.3 / 64.0 | 60.5 / 60.0 | 60.1 / 59.9 | 60.1 / 59.9 |
| TruthfulQA | 42.6 / 48.2 | 50.1 / 45.5 | 51.5 / 50.3 | 36.7 / 39.8 | 35.6 / 41.1 | 35.9 / 37.9 |
| GSM | 66.0 / 61.5 | 60.0 / 64.0 | 62.0 / 67.5 | 41.0 / 41.5 | 40.5 / 42.5 | 48.5 / 47.5 |
| BBH | 59.1 / 58.9 | 60.3 / 61.8 | 59.7 / 59.0 | 55.1 / 53.6 | 55.1 / 55.3 | 54.2 / 55.6 |
| TydiQA | 59.4 / 64.8 | 63.7 / 67.1 | 64.3 / 66.1 | 57.3 / 56.5 | 56.0 / 56.4 | 58.9 / 59.3 |
| Average | 58.1 / **59.5** | 59.7 / **60.6** | 60.2 / **61.4** | 50.1 / **50.3** | 49.5 / **51.0** | 51.5 / **52.0** |

**Score curation improves rating robustness** Furthermore, we explore the impact of score curation using different rating models. We compare the average performance results of $DS^2$ between without and with score curation in Figure 7 (Right). The base model is LLaMA-3.1-8B. For convenience, Figure 7 also demonstrates the maximum performance gap across three rating models under different data sizes. Notably, it is evident that with score curation, the average performance across rating models is more stable and shows improvement.

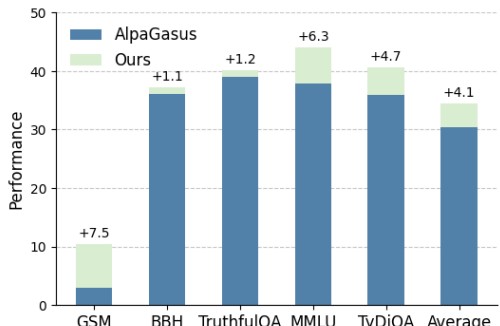

| | Performance gap ↓ |
|---|---|
| Data scale | w/o curation / w curation |
| 2.5k | 2.40 / **1.0** |
| 5k | 3.83 / **1.20** |
| 10k | 1.76 / **0.90** |
| 20k | 1.73 / **0.20** |
| 40k | **1.44** / 1.63 |
| Average | 1.60 / **0.70** |

Figure 7: **Left:** Apples-to-apples comparison with AlpaGasus using `LLaMA-2-7B` (base) on 9k samples from Alpaca subset (52k). **Right:** Maximum performance gap across different data scales.

### 6.3 APPLES-TO-APPLES COMPARISON WITH ALPAGASUS

To highlight $DS^2$'s superiority, we replicate AlpaGasus's settings for a fair apples-to-apples comparison. More details are in Appendix G.6. Using GPT-4o-mini for consistency, Figure 7 (Left) demonstrates that $DS^2$ significantly outperforms AlpaGasus with an improvement of **15%** in average, despite relying on a weaker rating model than AlpaGasus's default GPT-4 rating model.

## 7 CONCLUSION

In this paper, we challenge traditional data scaling laws in instruction tuning by introducing $DS^2$, a novel data selection pipeline that curates LLM-rated quality scores to improve data efficiency. Through the systematic exploration of error patterns in LLM-rated data quality scores, we developed a score curation mechanism to correct inaccuracies and enhance the effectiveness of selected data. Empirically, $DS^2$– using only 3.3% of the original data – outperforms training on the full dataset (300k samples) and even exceeds the performance of the human-aligned dataset "LIMA" with the same sample size (1k samples). This demonstrates that smaller, high-quality datasets can achieve superior results by avoiding performance drops caused by low-rated or redundant data, revising the traditional scaling laws that suggest more data is always better. By curating LLM-driven rating scores, $DS^2$ not only improves data efficiency, but also offers a cost-effective alternative to large-scale datasets and human annotations. Our results highlight the importance of data quality over quantity in instruction tuning and show how score curation can mitigate LLM biases, leading to improved model alignment and downstream performance. In conclusion, this work underscores the need to rethink data scaling laws in light of more efficient, curated data selection methods.

ACKNOWLEDGMENT

J. Pang and Y. Liu are partially supported by the National Science Foundation (NSF) under grants IIS-2007951, IIS-2143895, and IIS-2416896. J. Pang and C. Qian are also partially supported by NSF Grants 2322919, 2420632, and 2426031.

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

# APPENDIX

## ORGANIZATION OF THE APPENDIX

- **Section A**: Illustrates the limitations of this work.
- **Section B**: Provides more details of prompt-based LLM rating systems including more details of the data pool and prompt template.
- **Section C**: Presents a warm-up binary example to illustrate how to derive the score transition matrix, and the algorithm details of our proposed data selection pipeline $\mathbf{DS}^2$. In Appendix C.3, we analyze the $k$-NN clusterability hypothesis in detail. Besides, several 2-NN samples are also provided to evaluate the $k$-NN clusterability hypothesis.
- **Section D**: Explores the impact of embedding models.
- **Section E**: Explores the impact of score curation on examples by analyzing the rated score distribution, subset distribution as well as the score transition matrix.
- **Section F**: Demonstrates training and evaluation details.
- **Section G**: Provides more experimental results, including more downstream task evaluations, LLM judging evaluation, exploring the curation impact on score-aware methods, comparison with LIMA, new combined baseline which concatenating high-rated examples across rating models.
- **Section H**: Analyzes the computational complexity and runtime.
- **Section I**: Explores the impact of diversity score used for data selection.
- **Section J**: Presents several wrongly-rated examples by three rating models used in this work.

## A  LIMITATIONS

While the proposed method demonstrates competitive performance compared to other baselines, we acknowledge that there are still potential limitations:

- **Sample-independent assumption**. The sample-independent assumption is critical for deriving the transition matrix $\mathbf{T}$ and the true score probability distribution $\mathbf{p}$. However, this assumption may be somewhat strong and could inevitably introduce certain data-specific errors. Exploring weaker assumptions, such as group-dependent approaches, could be a valuable direction for future research.
- $k$-**NN clusterability**. The $k$-NN clusterability hypothesis implies that similar embedding vectors should correspond to the same rating score or class, a characteristic commonly leveraged in image classification tasks. However, in text-related tasks, highly similar texts can convey opposite semantic meanings due to subtle differences, such as a single word change. To address this challenge, powerful embedding models are essential to accurately distinguish these subtle differences and effectively capture the underlying semantic meaning.
- **Model scale**. Our experiments are primarily conducted on pre-trained models at the 7B/8B scale. It remains uncertain how well the method would perform on larger-scale pre-trained models.
- **Rating models**. Due to cost considerations, we use the more affordable GPT-4o-mini to generate GPT-level scores. It is unclear whether the score curation mechanism works for more powerful GPT models (e.g., GPT-4 or GPT-o1).

## B  PROMPT-BASED LLM RATING SYSTEMS

### B.1  DATA POOL

The data pool used in this work consists of five proceed datasets, which originate either from human annotations or generated by powerful LLMs. More details about these datasets are provided in Table 6. In particular, these datasets vary in format, quality, prompt length, and target tasks, demonstrating the diversity of our basic data pool. For convenience, we standardize the format of these datasets by using the "TULU" template format introduced by Wang et al. (2023). The "TULU" template consists of two main tags **<|User|>** and **<|Assistant|>**, reflecting the respective roles of the user and the assistant.

Table 6: Details of training datasets used in this work. WizardLM and Flan_v2 are sampled to 100K to match the dataset size. We report the average number of conservation turns ($\bar{N}_{\text{rounds}}$), average length of prompts ($\bar{L}_{\text{prompt}}$), average length of response ($\bar{L}_{\text{response}}$).

| Datasets | Sourced from | # Data size | Data quality | $\bar{N}_{\text{rounds}}$ | $\bar{L}_{\text{prompt}}$ | $\bar{L}_{\text{response}}$ |
|---|---|---|---|---|---|---|
| FLAN V2 | Human-generated instruction | 100K | Normal | 1.0 | 304.1 | 27.7 |
| OPEN-ASSISTANT 1 | human-generated instruction | 33K | Both | 1.6 | 32.3 | 189.1 |
| WIZARDLM | ChatGPT-generated instruction | 100K | High | 1.0 | 122.3 | 352.5 |
| DOLLY | Human-generated instruction | 15K | Normal | 1.0 | 99.5 | 79.3 |
| STANFORD ALPACA | Generated w/ Davinci-003 | 52K | Normal | 1.0 | 23.5 | 56.4 |

## B.2 QUALITY-BASED PROMPT TEMPLATE

The prompt template used in this work across various rating models is presented as follows. Our prompt template mainly accesses the data quality based on three criteria including rarity, complexity, and informativeness. For clarity and convenience, we adopt a JSON format to better capture the evaluation scores, following the LLaMA-3.1 template[2], as shown in Table B.2,.

---

**Prompt Template for LLM Rating**

**<System Prompt>**: As a data quality estimator, your task is to assess the quality of the data sample based on the criteria: Rarity, Complexity, and Informativeness. Please rate the sample on a scale from 1 to 10 for each criterion, and return an overall rating on a scale from 1 to 10, where a higher score indicates a higher level of quality. Ensure that the ratings are not overly concentrated around a specific score. If multiple samples have similar qualities, consider spreading the scores more evenly to reflect subtle differences.

**<User Prompt>**: Please carefully evaluate the following data sample and return the integral evaluation scores using the JSON format:
```
{"Rarity": <number, 1-10>,
    "Complexity": <number, 1-10>,
    "Informativeness": <number, 1-10>,
    "Overall rating": <number, 1-10>}
```

Instruction: [Instruction]
Input: [Input]
Response: [Response]

---

**Rated score rescaling**  Initially, to capture the subtle differences between data samples, we first prompt the LLMs to rate them on a continuous integer scale $\{1, 2, \cdots, 10\}$. Intuitively, a lower score indicates that the data sample is of lower quality. To simplify the score distribution, we first merge the lower scores in $\{1, 2, 3, 4\}$ and the higher scores in $\{9, 10\}$, resulting in a new scale of $\{4, 5, \cdots, 9\}$. For ease of convenience, we then shift this scale down to $\{0, 1, \cdots, 5\}$. Note that we focus primarily on high-rated samples in LLM ratings, so merging low-rated examples would not affect the overall performance and is more convenient for analyzing score errors in Section 3.2. Directly rating samples on a small scale of $\{0, 1, \cdots, 5\}$ seems more convenient but fails to capture the subtle difference between samples, especially among higher-rated samples. Meanwhile, this commonly leads to the issue where most of the samples are rated as 3. Starting with a larger scale and then narrowing it down allows LLMs to distinguish subtle quality differences in mid-rated samples better, improving performance.

## C DATA SELECTION PIPELINE $\mathbf{DS}^2$

### C.1 WARM-UP OF DERIVING SCORE TRANSITION MATRIX: A BINARY EXAMPLE

For a gentle start, let us consider a binary case ($K = 2$) with two types of scores $\{0, 1\}$. Here, $y$ represents the ground-truth score, while $\tilde{y}$ denotes the observed noisy score. We define the error rates (transition probabilities) as $e_{01} := \boldsymbol{T}(0, 1) := \mathbb{P}(\tilde{y} = 1 \mid y = 0)$ and $e_{10} := \boldsymbol{T}(1, 0) := \mathbb{P}(\tilde{y} = 0 \mid y = 1)$. According to the $k$-NN clusterability definition, similar embeddings are expected to

---

[2]https://www.llama.com/docs/model-cards-and-prompt-formats/llama3_1/

belong to the same category. Specifically, we focus on 2-**NN clusterability** in this work, meaning that the scores for the three samples within a 2-NN cluster should be identical, i.e., $y_1 = y_2 = y_3 = y$. Several target samples as well as their 2-NN samples are provided in Table 9. Note that the probabilities of the ground-truth score $p_i = \mathbb{P}(y = i), \forall i \in [K]$ also remain unknown. To estimate the exact values of the error rates $e_{01}$ and $e_{10}$, the high-level idea is to leverage higher-order consensus among $k$-NN cluster's scores, as outlined below.

- **First-order Concensuses**: We have

$$\mathbb{P}(\tilde{y}_1 = k) := \sum_{i \in [K]} \mathbb{P}(\tilde{y}_1 = k \mid y_1 = i), \forall k \in [K]$$

Then, we can obtain two first-order equations:

$$\mathbb{P}(\tilde{y}_1 = 0) := p_0(1 - e_{01}) + (1 - p_0)e_{10}$$

$$\mathbb{P}(\tilde{y}_1 = 1) := (1 - p_0)(1 - e_{10}) + p_0 e_{01}$$

- **Second-order Concensuses**: We have

$$\mathbb{P}(\tilde{y}_1 = k, \tilde{y}_2 = k') \stackrel{(a)}{=} \sum_{i \in [K]} \mathbb{P}(\tilde{y}_1 = k, \tilde{y}_2 = k' \mid y_1 = i, y_2 = i)\mathbb{P}(y_1 = i)$$

$$\stackrel{(b)}{=} \sum_{i \in [K]} \mathbb{P}(\tilde{y}_1 = k \mid y_1 = i)\mathbb{P}(\tilde{y}_2 = k' \mid y_2 = i)\mathbb{P}(y_1 = i), \forall k, k' \in [K]$$

where equality (a) holds due to the 2-NN clusterability and equality (b) holds because of the conditional independence between $\tilde{y}_1$ and $\tilde{y}_2$ based on their ground-truth score. Four second-order equations can be derived, e.g.,

$$\mathbb{P}(\tilde{y}_1 = 0, \tilde{y}_2 = 0) := p_0(1 - e_{01})^2 + (1 - p_0)e_{10}^2,$$

$$\mathbb{P}(\tilde{y}_1 = 1, \tilde{y}_2 = 1) := (1 - p_0)(1 - e_{10})^2 + p_0 e_{01}^2$$

- **Third-order Concensuses**: We have

$$\mathbb{P}(\tilde{y}_1 = k, \tilde{y}_2 = k', \tilde{y}_3 = k'') := \sum_{i \in [K]} \mathbb{P}(\tilde{y}_1 = k, \tilde{y}_2 = k', \tilde{y}_3 = k'' \mid y_1 = i, y_2 = i, y_3 = i)\mathbb{P}(y_1 = i)$$

Similarly, from different combinations of $\tilde{y}_1, \tilde{y}_2, \tilde{y}_3$, we have eight third-order equations, e.g.,

$$\mathbb{P}(\tilde{y}_1 = 1, \tilde{y}_2 = 1, \tilde{y}_3 = 1) := (1 - p_0)(1 - e_{10})^3 + p_0 e_{01}^3$$

Given the known score probability information $\mathbb{P}(\tilde{y}_1 = k)$, $\mathbb{P}(\tilde{y}_1 = k, \tilde{y}_2 = k')$ and $\mathbb{P}(\tilde{y}_1 = k, \tilde{y}_2 = k', \tilde{y}_3 = k'')$, we can utilize the above equations to derive the unknown ground truth score probability $p_0$ and error rates $e_{01}, e_{10}$. From these error rates, the transition matrix $\boldsymbol{T}$ can then be determined. For the entire dataset, we summarize the score probability information across all 2-NN clusters to derive the score transition matrix.

## C.2 ALGORITHM DETAILS

We provide the algorithm details of our proposed data selection pipeline in Algorithm 1.

## C.3 KNN CLUSTERABILITY HYPOTHESIS ANALYSIS

In this paper, the k-NN clusterability hypothesis is very crucial, which is based on the assumption that embeddings capture semantic and contextual similarity for textual data, which often correlates with quality and correctness. Similar to image classification tasks, these high-dimensional representations map semantically similar texts to nearby points in the vector space while positioning dissimilar texts farther apart, enabling clustering that aligns with classification categories. However, there may be a potential concern that samples with subtle token-level differences can yield different scores due to variations in correctness (the key factor). In this section, we will delve deeper into the practicality of the $k$-NN clusterability hypothesis for the following two reasons.

Firstly, our scoring approach considers not just correctness but also overall quality metrics such as rarity and informativeness, as outlined in our prompt template. This helps mitigate the influence of correctness alone on the final score. Additionally, we evaluate quality on a granular scale (e.g.,

---

**Algorithm 1** Proposed Data Selection Pipeline $\textbf{DS}^2$

---

1: **Input:** `Dataset` $D$, `EmbeddingModel`, `RawScores`, `TargetSize` $M$
2: **Output:** Selected subset $D^*$

3: **procedure** MODELING SCORE TRANSITION MATRIX(`Dataset`, `EmbeddingModel`)
4:     **Step-1:** Encode sample tuple and estimate score transition matrix
5:     `features` $\boldsymbol{x} \leftarrow$ ENCODING(`Dataset`, `EmbeddingModel`)
6:     `ConsensusInfo` $\leftarrow k$-NN STATISTICS INFO(`RawScores`)
7:     `T_Est` $\leftarrow$ ESTIMATETRANSITIONMATRIX(`ConsensusInfo`)         $\triangleright$ Consensuses Equation
8: **end procedure**

9: **procedure** SCORE CURATION MECHANISM(`Dataset`, `EmbeddingModel`)
10:     **Step-2:** Identify and curate misrated samples
11:     `CosSimilarityScores` $\leftarrow$ SIMILARITYSCORE($k$-`NNscores`, `RawScores`)
12:     `ErrorThreshold` $\leftarrow$ THRESHOLD(`DataSize`, `T_Est`)         $\triangleright$ Bayesian Rules
13:     `MisratedSamples` $\leftarrow$ SCORES RANKING(`CosSimilarityScores`, `ErrorThreshold`)
14:     `ConfidenceProbs` $\leftarrow$ IMBALANCERESCALING(`MisratedSamples`)
15:     `CuratedScores` $\leftarrow$ SCORECURATION(`MisratedSamples`, `ConfidenceProbs`)
16: **end procedure**

17: **procedure** LONG-TAIL SCORING(`Dataset`, `EmbeddingModel`)
18:     **Step-3:** Calculate the long-tail scores of examples based on $k$-NN distance
19:     **for** each sample's feature $\boldsymbol{x}_n$ in $D$ **do**
20:         `LongTailScores` $\leftarrow$ SIMILARITYSCORE(`feature` $\boldsymbol{x}_n$, `features` $\boldsymbol{x}$)         $\triangleright k$-NN Based
21:     **end for**
22: **end procedure**

23: **procedure** DATA SELECTION(`Dataset`, `EmbeddingModel`)
24:     **Step-4:** Leverage curated scores and long-tail scores to derive the selected subset $D^*$.
25:     $D_i \leftarrow$ GROUPING(`CuratedScores`)         $\triangleright i$ represents the score for each group
26:     **for** score $i$ in $\{5, 4, \cdots, 0\}$ **do**         $\triangleright$ Prioritize high-rated samples
27:         Sort $D_i$ by `LongTailScores` in descending order
28:         $D_i^* \leftarrow$ SELECTTOP($D_i$)         $\triangleright$ Select Top $M - |D^*|$ samples
29:         $D^* \leftarrow D^* \cup D_i^*$
30:         **if** $|D^*|$ equals to $M$ **then**
31:             **break**
32:         **end if**
33:     **end for**
34:     **Return** $D^*$
35: **end procedure**

---

$\{0, 1, \cdots, 10\}$, later compressed to $\{0, 1, \cdots, 5\}$) to reduce potential score discrepancies further. We provide randomly selected examples along with their 2-NN samples to demonstrate the validity of k-NN clusterability in our data pool, shown in Table 9. Moreover, we constructed specific examples where the raw LLM scores and the calculated embedding cosine similarity scores consistently align, confirming the correctness of the kNN clusterability hypothesis.

Secondly, the consensus vectors rely on the average probabilities across all 2-NN clusters, allowing statistical information from the remaining samples to mitigate corruption caused by a small number of violations. As a result, our method can tolerate a proportion of k-NN violations. Intuitively, prior work (Zhu et al., 2021) has demonstrated that even in image classification tasks, where **20%** of data samples violate the k-NN clusterability hypothesis, its method still outperforms other baselines. Empirically, our experimental results support this claim. Furthermore, due to the unavailability of ground-truth scores, it is infeasible to conduct experiments to explicitly detect such violations.

Here, we evaluate k-NN clusterability by examining the distribution of **average score gaps**, which measures the score difference within one $k$-NN cluster. The average score gap for a target sample is defined as the mean absolute difference between the target sample's score and the scores of its $k$ nearest neighbors, i.e.,

$$\text{Average score gap} = \text{Mean}(|\text{target samples score - kNN sample's score}|).$$

In our work, we focus on **2-NN clusterability** and frame our analysis within this context. Specifically, for each 2-NN cluster, we consider a target sample and its two nearest neighbors. For example,

given a 2-NN cluster with the score tuple: (target sample: 1, kNN sample 1: 2, kNN sample 2: 3), the score gap is calculated as: Average score gap $= \frac{|1-2|+|1-3|}{2} = 1.5$.

Table 7 summarizes the statistical distribution of score gaps across all 2-NN clusters. For a clearer visualization of score gap proportions with and without score curation, we further provide Figure 8.

Table 7: Average score gap statistical information of all 2-NN clusters from our data pool. We divide the score gap into five groups and outline the proportion of data in each.

| Curation | Model | Score Gap (0.0–1.0) (%) | Score Gap (1.5) (%) | Score Gap (2.0) (%) | Score Gap (>2.0) (%) |
|---|---|---|---|---|---|
| w/o Curation | GPT | 81.0 | 12.0 | 4.9 | 2.1 |
| w/o Curation | LLaMA | 58.3 | 18.0 | 12.2 | 11.5 |
| w/o Curation | Mistral | 70.2 | 16.5 | 8.1 | 5.4 |
| w/ Curation | GPT | 82.5 | 10.9 | 4.5 | 1.7 |
| w/ Curation | LLaMA | 78.8 | 9.4 | 7.3 | 4.1 |
| w/ Curation | Mistral | 80.5 | 10.8 | 5.6 | 4.3 |

From Table 7, we observe that **without score curation**, GPT has a higher proportion of samples in the 0.0–1.0 score gap range (81.0%) compared to Mistral (70.2%) and LLaMA (58.3%). This reveals that more powerful rating models, such as GPT, tend to exhibit smaller average score gaps, which aligns more closely with the concept of **k-NN clusterability** and contributes to improved performance.

Moreover, when comparing the settings **with and without score curation**, we observe that all three rating models show an increased proportion of samples in the 0.0–1.0 score gap range after score curation. Table 8 summarizes this comparison, including the corresponding average performance on LLM Leaderboard tasks. Therefore, these results demonstrate the validity of the proposed k-NN clusterability hypothesis.

Table 8: The proportion of samples in the 0.0–1.0 score gap range both with and without score curation for each rating model. For comparison, the corresponding average performance on LLM Leaderboard tasks is included in parentheses.

| Rating Model | Score Gap w/o Curation (Avg. Performance) | Score Gap w/ Curation (Avg. Performance) |
|---|---|---|
| GPT | 81.0% (60.2) | 82.5% (61.4) |
| LLaMA | 58.3% (59.2) | 78.8% (60.2) |
| Mistral | 70.2% (60.7) | 80.5% (61.1) |

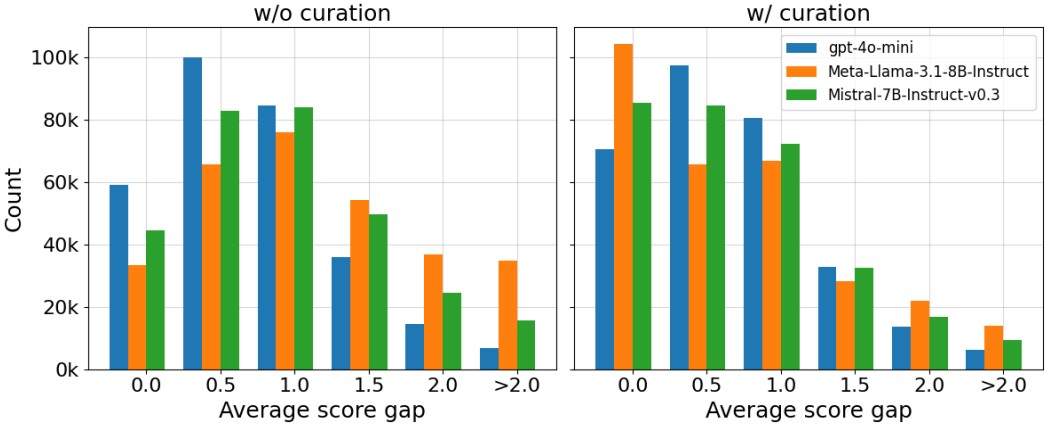

Figure 8: Average score gap statistical information of 2-NN clusters from our data pool. The average score gap for each target sample is defined as the average absolute score difference between the target sample and its 2-NN samples.

Table 9: Random selected target samples and their two nearest neighbors (2-NN) from our data pool. These examples are provided to evaluate the KNN clusterability, checking that similar embedding vectors correspond to the same category. The LLM rating score is from GPT-4o-mini, not the ground truth score. The ground truth score should be judged manually based on individual evaluation.

| Target sample | 1st nearest sample | 2nd nearest sample |
|---|---|---|
| <\|**Human**\|>: Pick any color from the rainbow.
<\|**Assistant**\|>: Purple.
**LLM rating score:** 1 | <\|**Human**\|>: Pick any color from the rainbow.
<\|**Assistant**\|>: Blue
**LLM rating score:** 0 | <\|**Human**\|>: List the colors of the rainbow
<\|**Assistant**\|>: Red, orange, yellow, green, blue, indigo, and purple
**LLM rating score:** 1 |
| <\|**Human**\|>: You need to complete the following task: Calculate 15% of the following number: 100
<\|**Assistant**\|>: 15% of 100 is 15.
**LLM rating score:** 1 | <\|**Human**\|>: Calculate 50% of 300.
<\|**Assistant**\|>: 50% of 300 is 150.
**LLM rating score:** 3 | <\|**Human**\|>: Calculate 15% of 500.
<\|**Assistant**\|>: 75
**LLM rating score:** 3 |
| <\|**Human**\|>: what is the color of mars?
<\|**Assistant**\|>: Rusty Red
**LLM rating score:** 2 | <\|**Human**\|>: what is the color of mars?
<\|**Assistant**\|>: Mars is called the Red Planet because of its reddish color.
**LLM rating score:** 5 | <\|**Human**\|>: Create a multiple choice question in which the answer is Mars.
<\|**Assistant**\|>: Which planet is known as the "Red Planet"?
A) Venus B) Jupiter C) Mars D) Uranus. Answer: C) Mars
**LLM rating score:** 1 |
| <\|**Human**\|>: What is the hexadecimal representation of the decimal number 74?
<\|**Assistant**\|>: 4A
**LLM rating score:** 1 | <\|**Human**\|>: Write a response that completes the request.Convert the number 18 to hexadecimal.
<\|**Assistant**\|>: The hexadecimal representation of 18 is 0x12.
**LLM rating score:** 0 | <\|**Human**\|>: Convert this hexadecimal value 0xAF to its equivalent decimal number
<\|**Assistant**\|>: 175
**LLM rating score:** 1 |
| <\|**Human**\|>: Use the word "scrumptious" in a sentence.
<\|**Assistant**\|>: The dinner was cooked to perfection and was absolutely scrumptious.
**LLM rating score:** 2 | <\|**Human**\|>: Delete two words from the following sentence to make it a valid sentence:
I ate the most delicious dinner ever.
<\|**Assistant**\|>: I ate most delicious ever.
**LLM rating score:** 2 | <\|**Human**\|>: Task: Edit the following sentence to make it more concise: "The food that we ate for the party was absolutely delicious."
<\|**Assistant**\|>: The food for the party was delicious.
**LLM rating score:** 2 |
| <\|**Human**\|>: List 3 applications of artificial intelligence.
<\|**Assistant**\|>: Some applications of Artificial Intelligence include natural language processing, facial recognition, and self-driving cars.
**LLM rating score:** 4 | <\|**Human**\|>: Name 3 applications for artificial intelligence.
<\|**Assistant**\|>: Three applications for artificial intelligence are natural language processing, computer vision, and facial recognition.
**LLM rating score:** 3 | <\|**Human**\|>: Name three artificial intelligence applications.
<\|**Assistant**\|>: Three artificial intelligence applications are: facial recognition software, natural language processing, and driverless cars.
**LLM rating score:** 1 |
| <\|**Human**\|>: Is there a negative or positive tone to this product review? Title: Worthless Review: This manual has little that applies to any Caddy and what may be in there is hard to locate. I found more on the internet than in this manual!
<\|**Assistant**\|>: Negative
**LLM rating score:** 3 | <\|**Human**\|>: Is there a negative or positive tone to this product review? Title: Did not use product. Review: I did not use the product because it had too many negative features to it. I do not know the process for returning items. I would like to return this item!
<\|**Assistant**\|>: Negative
**LLM rating score:** 4 | <\|**Human**\|>: Is there a negative or positive tone to this product review? Title: Terrible. review: this product is already displaying problems, which probably mean that i will be returning it soon for a replacement.
<\|**Assistant**\|>: Negative
**LLM rating score:** 2 |
| <\|**Human**\|>: Is there a negative or positive tone to this product review? Title: Medium is a little small. Review: Great thumb picks with solid feel. These seem to run on the small side and fit my thumb "snugly" but fortunately not too tight.
<\|**Assistant**\|>: Positive
**LLM rating score:** 1 | <\|**Human**\|>: Is there a negative or positive tone to this product review? Title: Comfortable Review: Would be a five but I was lucky to order a size bigger then usual and they fit ok! Could even be a little bigger but really comfortable!!
<\|**Assistant**\|>: Positive
**LLM rating score:** 1 | <\|**Human**\|>: Q: Is there a negative or positive tone to this product review? Title: A short cut Review: Great knives!!!! Work great and are absolutely beautiful. Own most of this line, and looking for more.
<\|**Assistant**\|>: Positive
**LLM rating score:** 2 |

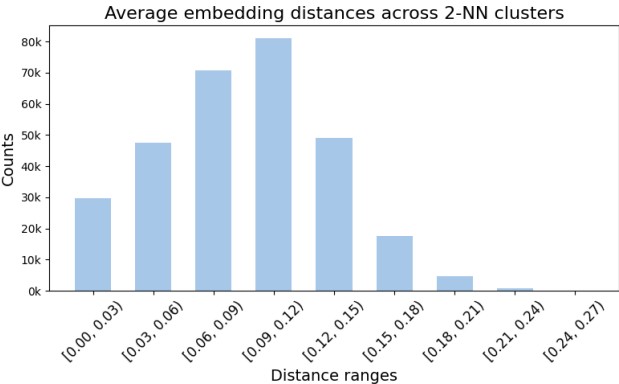

Figure 9: Average embedding distances across 2-NN clusters from our data pool. The embedding model is `BAAI/bge-large-en`.

## D    EXPLORING THE IMPACT OF EMBEDDING MODELS

By default, we use the newly released open-source model BGE as the embedding model throughout this paper. To explore the impact of embedding models, we adopt a popular alternative SetenceBERT (Reimers, 2019) to encode data samples. The score transition matrix across various rating models in the SetenceBERT embedding space is provided in Figure 10. Compared to Figure 3 in the BGE embedding space, we can observe that the impact of embedding space is limited, the choice of embedding model does not significantly affect the error patterns produced by LLMs.

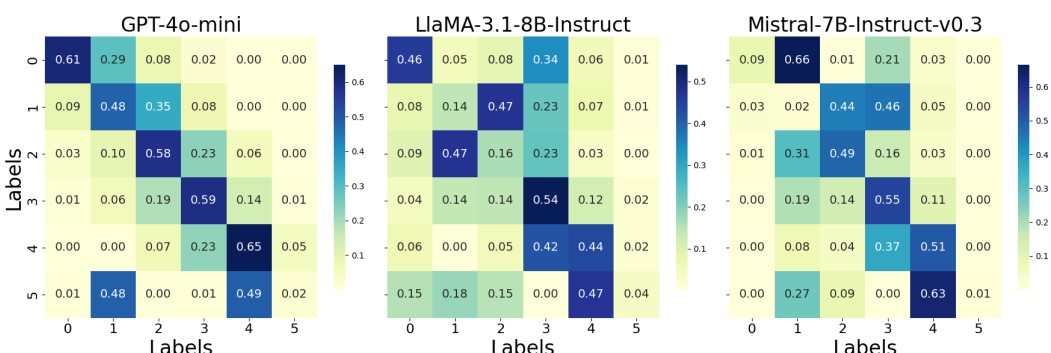

Figure 10: Score transition matrices across various rating models in the `SentenceBERT` embedding space.

## E    EXPLORING THE IMPACT OF SCORE CURATION ON EXAMPLES

### E.1    IMPACT OF SCORE CURATION ON DISTRIBUTION

**Rated score distribution between without and with curation**    Here, we compare the rated score distribution between without and with score curation, as shown in Figure 11. We observe a decrease in the number of high-rated examples, while the number of samples with a rating of 3 has increased significantly. The rationale behind this is that our score curation mechanism is based on $k$-NN statistical information. As a result, given the imbalanced distribution of rated scores, samples with a rating of 5 are rare and are inevitably drawn toward the majority rating of 3. Therefore, the results in Figure 11 also highlight the importance of confidence probability proposed in Section 4.

**Subset distribution of selected examples**    Recall that the data pool is constructed by five subsets. Here, we summarize the statistical information of 10K samples generated by $\mathbf{DS}^2$, focusing on the proportion of subsets. We can observe that 60%-70% of selected examples are from Wizardlm. The observation corresponds to the differences in data quality across five subsets summarized in Table 6.

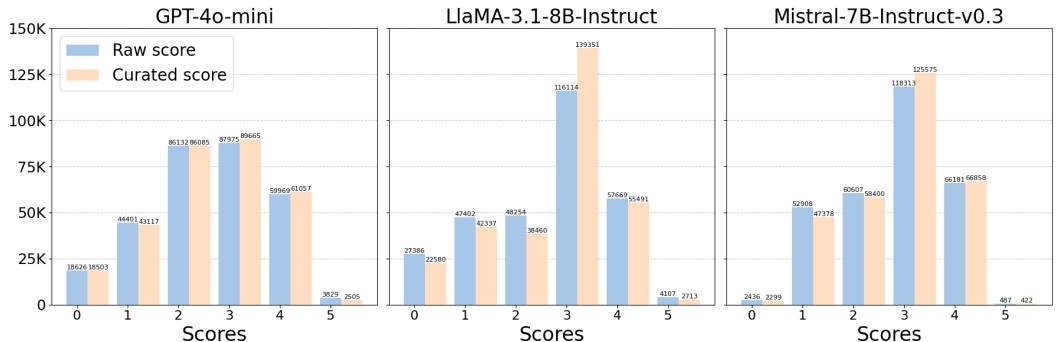

Figure 11: Comparison of rated score distribution between without and with score curation.

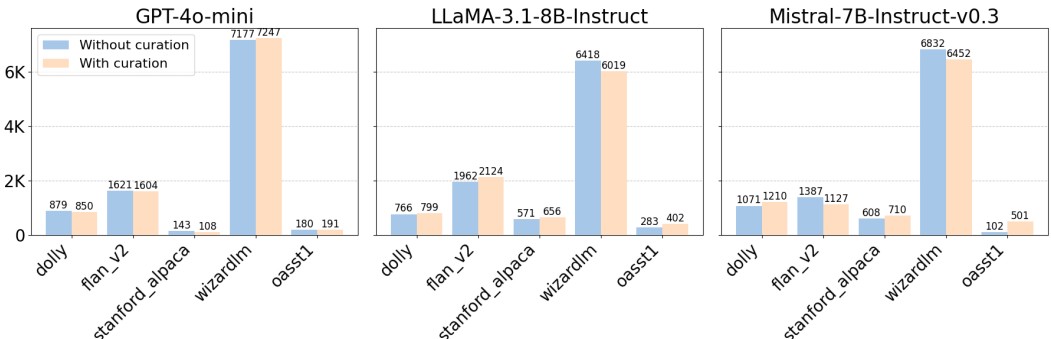

Figure 12: Subset distribution proportion within 10K samples generated by $\mathbf{DS}^2$.

## E.2 IMPACT OF SCORE CURATION ON SCORE ERRORS

Instead of the impact of score curation on final performance, we are also interested in the impact of score curation on the detected score transition matrix. Figure 13 illustrates the error pattern of different rating models after applying score curation. In comparison to the results without applying score curation illustrated in Figure 3, the improvements are remarkable. Our score curation mechanism can significantly reduce the probability of incorrect score transition in the matrices.

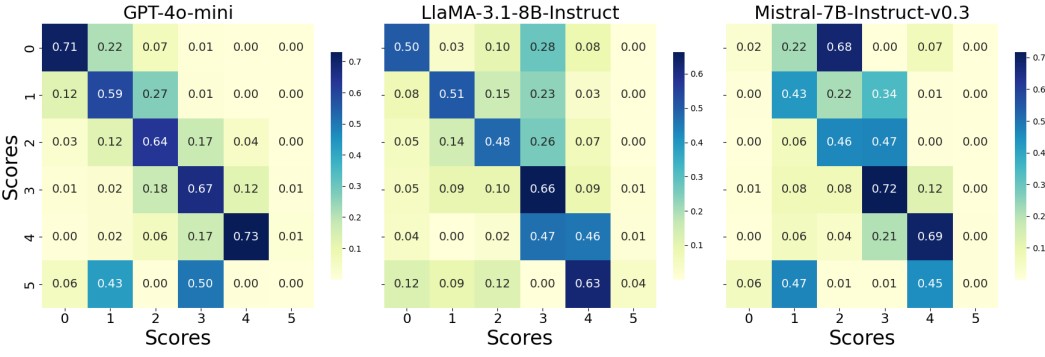

Figure 13: Score transition matrices comparisons across different rating models with score curation.

## F SETUP DETAILS

**Training details** In our experiments, we fine-tune 7B and 8B models using four or eight NVIDIA Tesla A100 GPUs. Following the experimental setup (Wang et al., 2023), for all experiments based on 7B/8B models, we consistently apply Lora (Hu et al., 2021) with a rank-size of 64 and a scaling factor of 16. Then, we set the overall batch size to 128, the learning rate at 1e-4, the training epochs

to 5, the dropout rate to 0.1, and a warm ratio of 0.03. The default maximum input length is 2048 tokens for all models.

**Evaluation details** In this paper, we select five tasks to conduct experiments for evaluation, consisting of MMLU, BBH, GSM, TydiQA, and TruthfulQA. The hyperparameter settings mainly follow recent work (Wang et al., 2023)'s. For ease of reproduction, we present some brief details.

- **MMLU** (Hendrycks et al., 2020): Following the setup of MMLU, we conduct all evaluations in the 0-shot setting without chain-of-thoughts (CoT).
- **GSM** (Cobbe et al., 2021): We evaluate fine-tuned models on a randomly selected subset with 200 samples from the original test set (1319 samples). In particular, we apply 8-shot in-context examples to simulate the CoT setting for reasoning.
- **BBH** (Suzgun et al., 2022): Given the official prompts provided in (Suzgun et al., 2022), we also apply 3-shot settings without CoT to make generations. Besides, we select 40 examples from each BBH sub-task.
- **TruthfulQA** (Lin et al., 2021): We prompt the fine-tuned models to generate answers for 818 TruthfulQA questions using the default QA prompt template with 6 in-context examples. Following the setting of (Wang et al., 2023), We apply two LLaMA-2-7B-based models for judging the generated responses' truthfulness[3] and informativeness[4]. Judge models will help to evaluate the truthful and informative rate of responses, respectively. We use 8-bit quantization to allow for efficient generation. Following (Lin et al., 2021), we finally take the Informative-Truthful Rate as our metric, which is calculated by the numerical product of the Informative and the Truthful Rate.
- **TydiQA** (Clark et al., 2020): This dataset is used to evaluate the model performance in answering multilingual questions across nine different languages. For each language, we select 100 examples. To help the models become familiar with the answer format, one in-context example is provided during testing. We report the average F1 score across various languages in this paper.

## G MORE EXPERIMENT RESULTS

### G.1 OPENLLM LEADERBOARD EVALUATION RESULTS

We conduct additional experiments to evaluate the performance of the OpenLLM leaderboard across different baselines, utilizing various base models such as Mistral-7B-v0.3 and LLaMA-2-7B-hf. Tables 10 and 11 present the results of the OpenLLM leaderboard using Mistral-7B-v0.3 and LLaMA-2-7B-hf as the base model, respectively. Both tables consistently demonstrate the effectiveness and superiority of our proposed pipeline $\mathbf{DS}^2$, following the previous claims provided in Secion 5.

### G.2 LLM JUDGE EVALUATION

To evaluate alignment performance across baselines, we utilize Vicuna-Bench to access the instruction-following ability (Chiang et al., 2023). Vicuna-Bench contains questions across nine domains, including generic, coding, math, and counterfactual. The judge model is GPT-4o-mini. Similarly, we present the final judge result in the typical "Win-Tie-Loss" rate form. For convenience, the judge prompt template as referenced in (Zheng et al., 2023) can be found in Table 12.

We compare all baselines, including our method against the full data baseline on Vicuna_Bench, as shown in Table 13. In particular, we conduct evaluations on two base models LLaMA-3.1-8B and Mistral-7B-v0.3. For score-aware baselines (AlpaGasus and Deita), we also compare them under three rating model settings. Notably, our method with curation outperforms almost all other baselines. What's more, in most cases, we can observe that the score curation step improves model performance by reducing the loss rate without compromising the original win rate.

### G.3 EXPLORING THE CURATION IMPACT ON OTHER SCORE-AWARE METHODS

Here, we present the curation impact on other score-aware methods, especially for Alpagasus and Deita under different rating model settings. The full experimental results can be found in Table 14.

---

[3]https://huggingface.co/allenai/truthfulqa-truth-judge-llama2-7B
[4]https://huggingface.co/allenai/truthfulqa-info-judge-llama2-7B

Table 10: Performance comparison on OpenLLM leaderboard. By default, the selected data size is 10K. Base model: `Mistral-7B-v0.3`. We highlight the best result in **boldface** and the second-best with underline.

| Models | MMLU (factuality) | TruthfulQA (truthfulness) | GSM (reasoning) | BBH (reasoning) | TydiQA (multilinguality) | Average |
|---|---|---|---|---|---|---|
| VANILLA BASE MODEL | 59.7 | 30.2 | 38.0 | 49.6 | 54.9 | 46.5 |
| COMPLETION LENGTH | 58.9 | 34.4 | 42.5 | 53.1 | 59.6 | 49.7 |
| PERPLEXITY | 59.8 | 40.3 | 36.0 | 48.9 | 57.4 | 48.5 |
| $k$-NN-10 | 58.3 | 41.7 | 43.5 | 54.1 | 53.4 | 50.2 |
| RANDOM SELECTION | 59.4 | 36.7 | 41.8 | 54.2 | 54.0 | 49.3 |
| LESS | 59.5 | 34.8 | 42.0 | 54.5 | 57.5 | 49.7 |
| FULL DATA (300K) | 60.0 | 43.5 | 43.5 | 52.5 | 53.4 | 50.6 |
| **Rating model: LLaMA-3.1-8B-Instruct** | | | | | | |
| ALPAGASUS | 59.9 | 36.4 | 39.0 | 52.6 | 56.3 | 48.8 |
| DEITA | 60.0 | 37.1 | 43.5 | 54.0 | 57.7 | 50.5 |
| OURS W/O CURATION | 60.0 | 37.2 | 45.0 | 53.5 | 54.5 | 50.0 |
| OURS | 59.7 | 37.8 | 48.5 | 54.4 | 55.2 | **51.1** |
| **Rating model: GPT-4o-mini** | | | | | | |
| ALPAGASUS | 60.5 | 36.7 | 41.0 | 55.1 | 57.3 | 50.1 |
| DEITA | 60.1 | 35.6 | 40.5 | 55.1 | 56.0 | 49.5 |
| OURS W/O CURATION | 60.1 | 35.9 | 48.5 | 54.2 | 58.9 | 51.5 |
| OURS | 59.9 | 37.9 | 47.5 | 55.6 | 59.3 | **52.0** |
| **Rating model: Mistral-7B-Instruct-v0.3** | | | | | | |
| ALPAGASUS | 59.5 | 35.6 | 46.0 | 55.7 | 52.1 | 49.8 |
| DEITA | 59.9 | 40.0 | 43.5 | 56.9 | 53.1 | 50.7 |
| OURS W/O CURATION | 59.5 | 37.9 | 46.5 | 55.8 | 57.2 | **51.4** |
| OURS | 59.5 | 40.3 | 48.5 | 53.0 | 55.9 | **51.4** |

Table 11: Performance comparison on OpenLLM leaderboard. By default, the selected data size is 10K. Base model: `LLaMA-2-7B-hf`. We highlight the best result in **boldface** and the second-best with underline.

| Model | MMLU (factuality) | TruthfulQA (truthfulness) | GSM (reasoning) | BBH (reasoning) | TydiQA (multilinguality) | Average |
|---|---|---|---|---|---|---|
| VANILLA LLaMA-2-7B | 41.9 | 28.4 | 6.0 | 38.3 | 35.7 | 30.1 |
| COMPLETION LENGTH | 42.4 | 36.4 | 1.5 | 36.8 | 33.9 | 30.2 |
| PERPLEXITY | 45.0 | 41.5 | 12.0 | 31.7 | 39.5 | 33.9 |
| $k$-NN-10 | 38.2 | 40.8 | 15.0 | 36.0 | 43.8 | 34.8 |
| RANDOM SELECTION | 44.7 | 41.8 | 14.0 | 37.9 | 40.8 | 35.8 |
| LESS | 44.3 | 38.2 | 18.0 | 35.2 | 46.3 | 36.4 |
| FULL DATA (300K) | 50.1 | 36.2 | 16.5 | 40.5 | 46.7 | 38.0 |
| **Rating model: llama-3.1-8B-Instruct** | | | | | | |
| ALPAGASUS | 45.1 | 41.2 | 18.0 | 35.6 | 39.8 | 35.9 |
| DEITA | 43.6 | 36.4 | 14.5 | 33.9 | 39.7 | 33.6 |
| OURS W/O CURATION | 45.4 | 39.7 | 15.0 | 35.5 | 42.1 | 35.5 |
| OURS | 44.9 | 44.9 | 14.0 | 38.3 | 44.8 | **37.4** |
| **Rating model: GPT-4o-mini** | | | | | | |
| ALPAGASUS | 45.3 | 41.0 | 14.5 | 37.0 | 45.3 | **36.6** |
| DEITA | 45.2 | 44.7 | 13.5 | 35.6 | 43.4 | 36.5 |
| OURS W/O CURATION | 42.0 | 39.5 | 15.0 | 38.1 | 46.1 | 36.1 |
| OURS | 40.2 | 43.8 | 13.5 | 38.9 | 46.5 | **36.6** |
| **Rating model: Mistral-7B-Instruct-v0.3** | | | | | | |
| ALPAGASUS | 42.3 | 41.9 | 16.0 | 34.1 | 41.6 | 35.2 |
| DEITA | 43.6 | 41.1 | 19.0 | 35.7 | 42.9 | 36.5 |
| OURS W/O CURATION | 46.0 | 48.6 | 15.0 | 35.2 | 43.7 | 37.7 |
| OURS | 40.8 | 50.9 | 15.0 | 37.9 | 45.5 | **38.0** |

## G.4 COMPARISON WITH HIGH-QUALITY HUMAN-ANNOTATED EXAMPLES: LIMA

In this section, we also utilize the original LIMA test set (300 samples) to compare the performance between LIMA (human annotation) and **DS**$^2$ (machine annotations). Similarly, we finetune two

Table 12: The prompt template used for GPT-4o judge evaluation from (Zheng et al., 2023)

**LLM Judge Prompt Template**

**System Prompt**:
You are a helpful and precise assistant for checking the quality of the answer.

**User Prompt**:
[Question]
[Assistant 1]: Assistant 1's Answer
[Assistant 2]: Assistant 2's Answer

We would like to request your feedback on the performance of two AI assistants in response to the user question displayed above. Please rate the helpfulness, relevance, accuracy, level of details of their responses. Each assistant receives an overall score on a scale of 1 to 10, where a higher score indicates better overall performance. Please first output a single line containing only two values indicating the scores for Assistant 1 and 2, respectively. The two scores are separated by a space. In the subsequent line, please provide a comprehensive explanation of your evaluation, avoiding any potential bias and ensuring that the order in which the responses were presented does not affect your judgment.

Table 13: Performance comparison with full data baseline on Vicuna_Bench. Base models: `LLaMA-3.1-8B` and `Mistral-7B-v0.3`. LLM judge model: GPT-4o-mini. $\widetilde{\text{Win}}$ represents the adjusted win rate, which equals the win rate plus half of the tie rate. We highlight the best result in **boldface** and the second-best with underline.

| Model | LLaMA-3.1-8B | | | | Mistral-7B-v0.3 | | | |
|---|---|---|---|---|---|---|---|---|
| | Win(%) | Loss(%) | Tie(%) | $\widetilde{\text{Win}}$(%) | Win(%) | Loss(%) | Tie(%) | $\widetilde{\text{Win}}$(%) |
| COMPLETION LENGTH | 55.5 | 32.5 | 12.0 | 61.5 | 61.3 | 25.0 | 13.8 | 68.1 |
| PERPLEXITY | 35.6 | 51.3 | 13.1 | 42.2 | 45.0 | 38.8 | 16.3 | 53.1 |
| $k$-NN-10 | 51.3 | 29.4 | 19.4 | 60.9 | 51.3 | 32.5 | 16.3 | 59.4 |
| RANDOM SELECTION | 33.1 | 45.0 | 21.9 | 44.1 | 46.3 | 35.0 | 18.8 | 55.6 |
| LESS | 35.0 | 51.3 | 13.8 | 41.9 | 36.3 | 48.8 | 15.0 | 43.8 |
| **Rating model: LLaMA-3.1-8B-Instruct** | | | | | | | | |
| ALPAGASUS | 50.6 | 28.8 | 20.6 | 60.9 | 57.5 | 27.5 | 15.0 | 65.0 |
| DEITA | 40.6 | 45.0 | 14.4 | 47.8 | 46.3 | 36.3 | 17.5 | 55.0 |
| OURS W/O CURATION | 56.3 | 30.0 | 13.8 | **63.1** | 55.0 | 30.0 | 15.0 | 62.5 |
| OURS | 53.8 | 27.5 | 18.8 | **63.1** | 63.8 | 22.5 | 13.8 | **70.6** |
| **Rating model: GPT-4o-mini** | | | | | | | | |
| ALPAGASUS | 67.5 | 18.8 | 13.8 | 74.4 | 73.8 | 10.3 | 15.9 | **81.7** |
| DEITA | 54.6 | 32.1 | 13.3 | 61.3 | 63.1 | 26.3 | 10.6 | 68.4 |
| OURS W/O CURATION | 70.4 | 19.6 | 10.0 | **75.4** | 67.5 | 22.5 | 10.0 | 72.5 |
| OURS | 63.8 | 20.0 | 16.3 | 71.9 | 65.0 | 20.0 | 15.0 | 72.5 |
| **Rating model: Mistral-7B-Instruct-v0.3** | | | | | | | | |
| ALPAGASUS | 48.8 | 22.5 | 28.8 | **63.1** | 55.0 | 28.8 | 16.3 | 63.1 |
| DEITA | 46.3 | 36.3 | 17.5 | 55.0 | 45.0 | 41.9 | 13.1 | 51.6 |
| OURS W/O CURATION | 51.7 | 33.8 | 14.6 | 58.9 | 61.9 | 25.0 | 13.1 | 68.4 |
| OURS | 51.3 | 31.3 | 17.5 | 60.0 | 62.5 | 20.0 | 17.5 | **71.3** |

base models (LLaMA-3.1-8B and Mistral-7B-v0.3) on 1k LIMA samples. The finetuned models are then directly compared with finetuned models using **DS**$^2$ selected examples at both 1k and 10k sample sizes. The experimental results for 1k and 10k settings are shown in Figure 14a and 14b, respectively. While **DS**$^2$ performs worse than LIMA in the 1k sample setting, it totally surpasses LIMA in the 10k setting, consistently demonstrating the superiority of **DS**$^2$. This lower performance at the 1k setting is expected, as LIMA has a natural advantage in a limited sample size scenario due to the IID nature of its training and test sets.

Table 14: Performance comparison between without and with score curation across all score-aware methods. Results are presented as (without curation **/** with curation). The selected base models are `LLaMA-3.1-8B` and `Mistral-7B-v0.3`.

| | **Rating Model: LLaMA-3.1-8B-Instruct** | | | | | |
| | **LLaMA-3.1-8B** | | | **Mistral-7B-v0.3** | | |
| | ALPAGASUS | DEITA | OURS | ALPAGASUS | DEITA | OURS |
|---|---|---|---|---|---|---|
| MMLU | 63.1 / 63.8 | 64.1 / 64.6 | 63.4 / 63.8 | 59.9 / 59.4 | 60.0 / 59.8 | 60.0 / 59.7 |
| TruthfulQA | 42.4 / 36.1 | 35.3 / 46.3 | 50.2 / 45.4 | 36.4 / 41.7 | 37.1 / 39.8 | 37.2 / 37.8 |
| GSM | 59.5 / 65.5 | 60.0 / 64.0 | 61.5 / 62.5 | 39.0 / 40.0 | 43.5 / 43.0 | 45.0 / 48.5 |
| BBH | 60.9 / 63.1 | 60.8 / 58.3 | 59.3 / 61.2 | 52.6 / 53.5 | 54.0 / 52.4 | 53.5 / 54.4 |
| TydiQA | 64.8 / 62.7 | 63.0 / 61.3 | 61.7 / 67.9 | 56.3 / 52.3 | 57.7 / 58.0 | 54.5 / 55.2 |
| Average | 58.1 / **58.2** | 56.6 / **58.9** | 59.2 / **60.2** | 48.8 / **49.4** | 50.5 / **50.6** | 50.0 / **51.1** |

| | **Rating Model: GPT-4o-mini** | | | | | |
| | **LLaMA-3.1-8B** | | | **Mistral-7B-v0.3** | | |
| | ALPAGASUS | DEITA | OURS | ALPAGASUS | DEITA | OURS |
|---|---|---|---|---|---|---|
| MMLU | 63.4 / 64.1 | 64.5 / 64.6 | 63.3 / 64.0 | 60.5 / 60.0 | 60.1 / 59.9 | 60.1 / 59.9 |
| TruthfulQA | 42.6 / 48.2 | 50.1 / 45.5 | 51.5 / 50.3 | 36.7 / 39.8 | 35.6 / 41.1 | 35.9 / 37.9 |
| GSM | 66.0 / 61.5 | 60.0 / 64.0 | 62.0 / 67.5 | 41.0 / 41.5 | 40.5 / 42.5 | 48.5 / 47.5 |
| BBH | 59.1 / 58.9 | 60.3 / 61.8 | 59.7 / 59.0 | 55.1 / 53.6 | 55.1 / 55.3 | 54.2 / 55.6 |
| TydiQA | 59.4 / 64.8 | 63.7 / 67.1 | 64.3 / 66.1 | 57.3 / 56.5 | 56.0 / 56.4 | 58.9 / 59.3 |
| Average | 58.1 / **59.5** | 59.7 / **60.6** | 60.2 / **61.4** | 50.1 / **50.3** | 49.5 / **51.0** | 51.5 / **52.0** |

| | **Rating Model: Mistral-7B-Instruct-v0.3** | | | | | |
| | **LLaMA-3.1-8B** | | | **Mistral-7B-v0.3** | | |
| | ALPAGASUS | DEITA | OURS | ALPAGASUS | DEITA | OURS |
|---|---|---|---|---|---|---|
| MMLU | 63.2 / 64.2 | 63.9 / 63.5 | 63.0 / 63.3 | 59.5 / 59.6 | 59.9 / 59.5 | 59.5 / 59.5 |
| TruthfulQA | 45.8 / 40.0 | 50.3 / 51.3 | 48.2 / 53.9 | 35.6 / 38.9 | 40.0 / 38.7 | 37.9 / 40.3 |
| GSM | 62.0 / 60.5 | 61.0 / 61.0 | 67.0 / 62.0 | 46.0 / 46.5 | 43.5 / 44.0 | 46.5 / 48.5 |
| BBH | 60.5 / 63.5 | 60.4 / 59.5 | 59.2 / 61.1 | 55.7 / 55.6 | 56.9 / 54.1 | 55.8 / 53.0 |
| TydiQA | 62.2 / 63.5 | 62.8 / 64.6 | 65.9 / 65.1 | 52.1 / 56.6 | 53.1 / 55.1 | 57.2 / 55.9 |
| Average | **58.7** / 58.3 | 59.7 / **60.0** | 60.7 / **61.1** | 49.8 / **51.4** | **50.7** / 50.3 | **51.4** / **51.4** |

## G.5 EXPLORING THE IMPACT OF CONCATENATING HIGH-RATED EXAMPLES ACROSS RATING MODELS

**Combined baseline** Here, we are also interested in the performance of concatenating samples from three rating models. We combined all high-rated samples with a score of 5, resulting in a subset of 8K samples. To reach a total of 10K samples, we added 2K samples from the data pool that were both rated 4 by all rating models. Compared to the results shown in Table 3 and Table 10, one can observe that the combined baseline still fails to achieve strong performance.

Table 15: Performance of COMBINED baseline on OpenLLM Leaderboard.

| | **Combined baseline** | |
| | **LLaMA-3.1-8B** | **Mistral-7B-v0.3** |
|---|---|---|
| MMLU | 64.2 | 59.6 |
| TruthfulQA | 41.7 | 37.1 |
| GSM | 62.5 | 43.5 |
| BBH | 61.9 | 51.0 |
| TydiQA | 60.8 | 53.1 |
| Average | 58.2 | 48.9 |

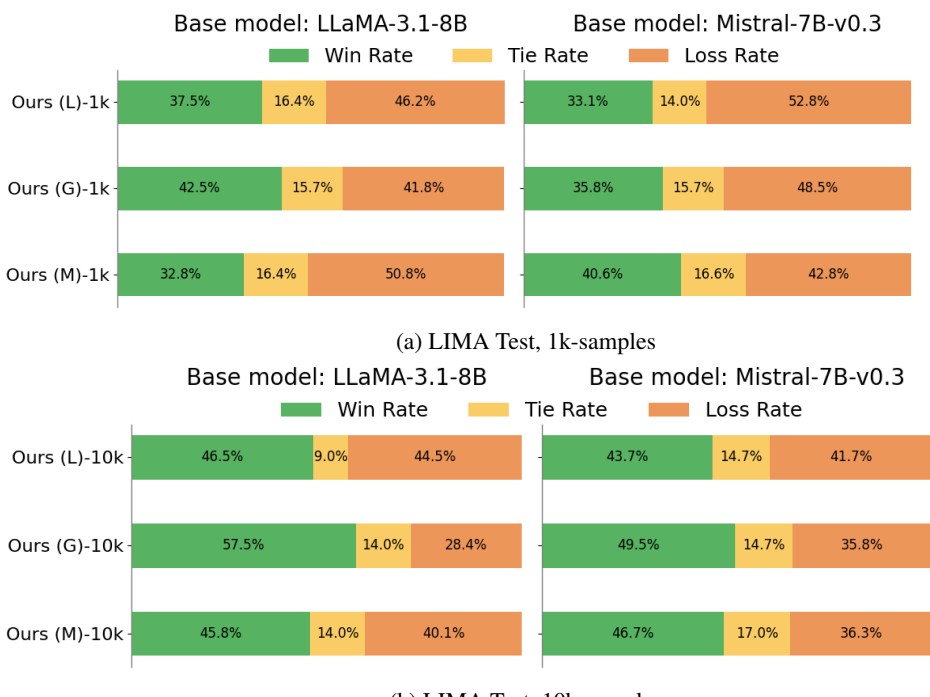

(a) LIMA Test, 1k-samples

(b) LIMA Test, 10k-samples

Figure 14: Performance of models fintuned on $DS^2$ (10k samples, machine-curated) v.s. LIMA (1k samples, human-curated). Evaluation set: LIMA (300 samples). We use the initial letter to denote the rating model, e.g., **Ours (L)** refers to our method with LLaMA-generated scores (**Ours (LLaMA)**).

### G.6 Apples-to-apples Performance Comparison with AlpaGasus

Note that the raw scores used in this work for AlpaGasus Chen et al. (2023) are generated with our prompt template. Our prompt template largely follows the format and criteria of Alpagasus (as the first rating prompt template), maintaining alignment with established standards. A significant improvement in our approach is using JSON format to return evaluation scores, allowing us to capture the scores accurately. This JSON formatting approach is inspired by the official LLama-3.1 chat template, as detailed in LLama-3.1 model documentation. We conduct experiments to compare our method with AlpaGasus under the same 4-bit quantization and LoRA settings, adhering closely to the same experimental configurations. The `AlpaGasus-2-7B-QLoRA` model originates from a related repository highlighted in the official AlpaGasus repository, with LLaMA-2-7B as the base model. The rating scores used in our method are generated from GPT-4o-mini, which is much weaker than GPT-4 used in AlpaGasus.

## H Computational Complexity

Table 16 summarizes the storage and GPU running time of our method as well as three representative baselines. The wall-clock running time is measured on a Microsoft Azure 8*A100 (80GB) GPUs cluster. Note that our score curation mechanism relies primarily on linear programming (LP), which runs exclusively on the CPU. As shown in the table, LLM rating systems are advantageous over the gradient-based method LESS in terms of both storage and runtime. Notably, compared to AlpaGasus and DEITA, our method avoids any significant computation costs on the GPU.

## I Exploring the Impact of Diversity Score

The importance of diversity on LLM data selection has been extensively explored by previous work Wang et al. (2023); Liu et al. (2023a); Wang et al. (2022). Note that our data pool is composed of five distinct subsets, each characterized by varying levels of complexity and diversity. The statistical analysis of diversity scores across subsets, as illustrated in Figure 15, confirms this. To evaluate

Table 16: Comparison of storage and running time.

| | Storage | Running Time | | | | Base Model Free | Validation Set |
|---|---|---|---|---|---|---|---|
| | | Rating/Gradient | Diversity Score | CPU-only Curation | Data Selection | | |
| LESS | 20GB | 66H | - | - | <1mins | No | Required |
| AlpaGasus | <10MB | 6H | - | - | <1mins | Yes | Not Required |
| DEITA | <10MB | 6H | 10 mins | - | <1mins | Yes | Not Required |
| Ours | <10MB | 6H | - | 25 mins | <1mins | Yes | Not Required |

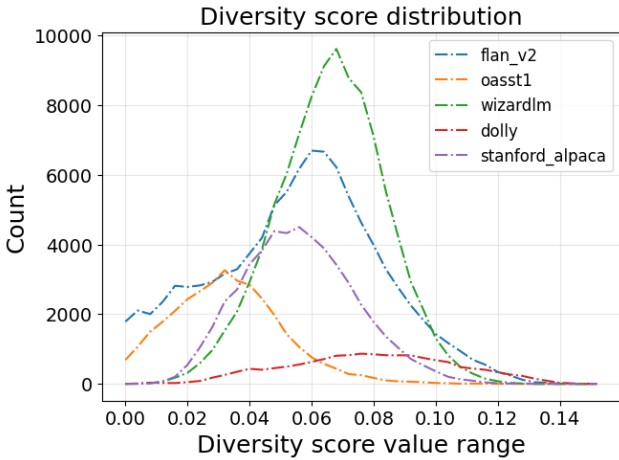

Figure 15: Subset diversity score distribution. The diversity score distribution across subsets demonstrates that the complexity and diversity are different.

the versatility of the diversity score, we further conduct additional contrast experiments here. In particular, we solely rank the samples of subsets based on the diversity score. Then, we select the Top-k and Bottom-k samples independently to construct datasets for LLM instruction finetuning, where k =10000. The corresponding performance results are presented in the following table. For cost considerations, we employ LLaMA-3.2-3B as the base model. The experimental settings are consistent with those outlined in our paper. From the table, it is evident that the diversity score is not universally effective across all datasets. To achieve better results, it should be complemented with other specific metrics, such as LLM rating scores.

Table 17: Performance comparison between low and high diversity score across different datasets. The base model is `LLaMA-3.2-3B`. **Bottom-k** (**Top-k**) refers to the samples with the lowest (highest) diversity scores, where $k = 10000$.

| Metric | Flan_v2 | | Wizardlm | | Stanford Alpaca | |
|---|---|---|---|---|---|---|
| | Bottom-k | Top-k | Bottom-k | Top-k | Bottom-k | Top-k |
| MMLU | 55.6 | 54.8 | 56.7 | 56.6 | 56.5 | 55.1 |
| BBH | 44.9 | 45.0 | 45.8 | 47.7 | 46.3 | 47.1 |
| GSM8K | 24.5 | 29.5 | 30.5 | 28.5 | 28.5 | 26.0 |
| TruthfulQA(MC2) | 38.6 | 41.7 | 46.6 | 48.1 | 40.2 | 40.6 |
| Tydiqa | 55.9 | 60.5 | 37.7 | 31.2 | 48.4 | 39.5 |
| Average | 43.9 | 46.3 | 43.5 | 42.4 | 44.0 | 41.7 |

## J MIS-RATED EXAMPLES

Here, we provide more wrongly-rated examples detected by our algorithms. For convenience, we highlight the raw (original) rated score and the curated score of each example.

Wrongly-rated example by LLaMA-3.1-8B-Instruct. Raw rated score: 5; Curated score:1

**\<User\>**: Here is some data about a restaurant: name = The Eagle, eatType = coffee shop, food = Fast food, customer rating = 3 out of 5, area = riverside, familyFriendly = yes, near = Burger King. Write a sentence that includes the above data about a restaurant
**\<Assistant\>**: The Eagle is a kid-friendly coffee shop that is located on the riverside near Burger King. The Eagle serves fast food and is moderate in pricing. Customers rate the coffee shop 3 out of 5.

Wrongly-rated example by LLaMA-3.1-8B-Instruct. Raw score: 5; Curated score:1

**\<User\>**: Based on the premise "i think it would too", can we conclude the hypothesis "I also think it would. " is true (see options)? Pick from: (i) yes; (ii) it is not possible to tell; (iii) no;
**\<Assistant\>**: (i).

Wrongly-rated example by GPT-4o-mini. Raw score: 5; Curated score:0

**\<User\>**:Premise: Island society has seen more change in the last twenty years than in the previous thousand. Hypothesis: Island society has been changing rapidly because of boats with powerful motors. Is the hypothesis entailed by the premise? Select from: 1. yes. 2. it is not possible to tell. 3. no. And the answer is:
**\<Assistant\>**: 2.

Wrongly-rated example by GPT-4o-mini. Intial score: 5; Curated score:1

**\<User\>**: De façon plus précise, si acc scc c et c représentent la consommation totale des agents avec et sans contraintes de crédit, respectivement, léquation de consommation globale devient c = c c c acc acc. Which language is this?
**\<Assistant\>**: French.

Wrongly-rated example by GPT-4o-mini. Raw score: 5; Curated score:2

**\<User\>**: Ask a question about Queen Victoria.
**\<Assistant\>**: Who did Victoria dislike so much she banned him from her presence once she became Queen?

Wrongly-rated example by Mistral-7B-v0.3-Instruct. Raw score: 5; Curated score:1

**\<User\>**: Add punctuation: Yahweh sent Jerubbaal and Bedan and Jephthah and Samuel and delivered you out of the hand of your enemies on every side and you lived in safety
**\<Assistant\>**: Yahweh sent Jerubbaal, and Bedan, and Jephthah, and Samuel, and delivered you out of the hand of your enemies on every side; and you lived in safety.

Wrongly-rated example by Mistral-7B-v0.3-Instruct. Raw score: 5; Curated score:1

**\<User\>**: Die Hausordnung ist Bestandteil der allgemeinen Geschäftsbedingungen. Translate this to English?
**\<Assistant\>**: The house rules of the Hotel are an integral part of the General Rules and Conditions.

> **Wrongly-rated example by Mistral-7B-v0.3-Instruct. Raw score: 5; Curated score:1**
>
> **<User>**: Generate a sentence that describes the following data: Alan Frew, occupation, Public speaking; Alan Frew, birthPlace, United Kingdom; Alan Frew, origin, Canada; Alan Frew, background, "solo singer"; Alan Frew, origin, Newmarket, Ontario. Sentence:
> **<Assistant>**: Andrew Frew was a solo singer who makes his living public speaking. Alan Frew was born in the UK but has Canadian origins coming from Newmarket, Ontario.

### J.1 ORIGINAL HIGH-RATED EXAMPLES VS. ORIGINAL LOW-RATED EXAMPLES

We provide several high-rated/low-rated examples from different datasets to illustrate the data quality, including Flan_v2, Open-Assistant1, Stanford Alpaca, Dolly, and WizardLM.

Table 18: Comparison of low-rated and high-rated samples in Dolly and Stanford Alpaca datasets

| Low-rated Dolly sample | High-rated Dolly sample |
|---|---|
| <\|**Human**\|>: Instruction: Below is an instruction that describes a task, paired with an input that provides further context. Write a response that appropriately completes the request.
**Instruction:** When was the 8088 processor released?
**Input:** The 8086 (also called iAPX 86) is a 16-bit microprocessor chip designed by Intel between early 1976 and June 8, 1978, when it was released. The Intel 8088, released July 1, 1979, is a slightly modified chip with an external 8-bit data bus (allowing the use of cheaper and fewer supporting ICs),[note 1] and is notable as the processor used in the original IBM PC design.
<\|**Assistant**\|>:
**Response**: The Intel 8088 processor was released July 1, 1979. | <\|**Human**\|>: Instruction: Below is an instruction that describes a task, paired with an input that provides further context. Write a response that appropriately completes the request.
**Instruction:** Who is the best surfer in the world?
**Input:** Robert Kelly Slater (born February 11, 1972) is an American professional surfer, best known for being crowned World Surf League champion a record 11 times. Slater is widely regarded as the greatest professional surfer of all time, and holds 56 Championship Tour victories. Slater is also the oldest surfer still active on the World Surf League, winning his 8th Billabong Pipeline Masters title at age 49.
<\|**Assistant**\|>:
**Response**: Kelly Slater |
| **Low-rated Stanford Alpaca sample** | **High-rated Stanford Alpaca sample** |
| <\|**Human**\|>: Instruction: Given a set of numbers, find the maximum value.
**Input:** Set: {10, 3, 25, 6, 16}
<\|**Assistant**\|>:
**Response**: The maximum value is 25. | <\|**Human**\|>: Instruction: Create a list of best practices that should be followed in this field.
**Field:** Healthcare information technology
<\|**Assistant**\|>:
**Response**: The following best practices should be followed in the healthcare information technology field:
1. Adopt a risk-based approach to cybersecurity, utilizing measures such as multi-factor authentication, encryption, and vulnerability assessments.
2. Develop clear guidelines and policies governing the use of technology.
3. Develop protocols to ensure the secure sharing of patient data.
4. Implement robust data processing and storage systems that adhere to industry regulations.
5. Train healthcare staff in both the technical and non-technical aspects of IT use.
6. Utilize professional IT staff for any maintenance or development tasks.
7. Regularly test and monitor IT systems for potential vulnerabilities. |

Table 19: Comparison of low-rated and high-rated samples from Flan_v2 and WizardLM datasets

| Low-rated Flan_v2 sample | High-rated Flan_v2 sample |
|---|---|
| <\|**Human**\|>: Triple: Alimentum eatType restaurant; Alimentum food Indian; Alimentum priceRange moderate; Alimentum area riverside; Alimentum familyFriendly no. What is a sentence that describes this triple?
<\|**Assistant**\|>: "Alimentum is an Indian restaurant located the the riverside area. It is moderately priced, but not kid friendly." | <\|**Human**\|>: I walked in here during an all day rain-a-thon in Pittsburgh (so rare, right?) on a Saturday. It was pretty busy, but all of the hustle and bustle was contained to the upstairs, which looked like a cool loft area. It wasn't loud either. The first thing I noticed is how high the ceiling was and I loved that. The art on the walls was a delight to look at too. I ordered something vegan with tofu (I'm not vegan by any means, but wanted to try it), and I got it delivered to me on a plate with two small toasted pieces of bread, lettuce, slices or bell pepper, and warmed tofu. There was something else on the bread that I don't remember, but I was really surprised by how delicious and satisfying this was. I am a fan. For my drink I got an iced vanilla latte and I have to say it's the best vanilla latte I've had around town. I ended up having two more throughout my time there...I just inhaled them. Seriously..so good. I came back on Sunday and had an iced mocha, which was just as good. This is my new favorite place for real!
Choose: was this review given positively or negatively?
<\|**Assistant**\|>: positive |

| Low-rated WizardLM sample | High-rated WizardLM sample |
|---|---|
| <\|**Human**\|>: Instruction: As an intermediate English learner at B1 level, I have been trying to familiarize myself with certain words like "captivity", "chew out" and "cuddly". Would you kindly ask me a question using any of these three terms, to help me better comprehend the usage and context of these words?
<\|**Assistant**\|>:
**Response**: Certainly! Here are three questions using each of the terms you mentioned:
1. Have you ever visited a zoo where animals are kept in captivity?
2. Has your boss ever chewed you out for making a mistake at work?
3. Do you prefer to have a pet that is cuddly and affectionate, or one that is independent and aloof? | <\|**Human**\|>: Instruction: As an archaeologist exploring the depths of an uncharted jungle, you come across a temple that appears to be a blend of Mayan and Roman architecture. The engravings on the walls and artifacts found within the temple suggest that the civilization that built it was a fusion of both cultures. Your task is to piece together the history and culture of this ancient civilization and unravel the mystery of how and why it came to be. Were these two cultures united in peace or war? Did they form a new civilization or remain separate? Let's excavate the truth behind this enigma.
<\|**Assistant**\|>:
**Response**: As you begin your exploration of the temple, you come across a set of glyphs on one of the walls that seem to represent both Mayan and Roman symbols. You take note of them and continue your journey deeper into the temple. Inside, you find a variety of artifacts that suggest a fusion of both cultures, including pottery, weapons, and clothing....

In the end, you realize that this ancient civilization was a testament to the power of cooperation and cultural exchange. The Mayan and Roman peoples came together in peace, combining their knowledge and skills to create something new and beautiful. Their legacy lives on in the unique artifacts and engravings that remain in the temple, waiting to be discovered and appreciated by future generations. |

Table 20: Comparison of low-rated and high-rated samples from Open-Assistant 1 dataset

| Low-rated Open-Assistant 1 sample | High-rated Open-Assistant 1 sample |
|---|---|
| <\|**Human**\|>: que actividades se supone que son comunes en una persona de 20 años?
<\|**Assistant**\|>: Las actividades más comunes entre personas de 20 años usualmente son estudiar, debido a la alta demanda de tareas en la universidad, así como, salir a divertirse, tomar alcohol eventualmente, o salir de fiesta, así mismo, se puede incluir hacer deporte, y jugar videojuegos.
<\|**Human**\|>: Las actividades que se supone que son comunes en una persona de 20 años varían dependiendo de muchos factores, como la cultura, el entorno, la educación y los intereses personales. Sin embargo, aquí hay algunas actividades que pueden ser comunes para muchas personas de 20 años:
• Estudiar o trabajar: Muchas personas de 20 años se encuentran en una etapa de transición entre la educación y el mundo laboral, por lo que pueden estar estudiando o trabajando en sus primeras carreras o empleos.
• Establecer relaciones: Las personas de 20 años a menudo están explorando su vida amorosa y estableciendo relaciones íntimas.
• Viajar: Muchas personas de 20 años aprovechan esta etapa para viajar y explorar el mundo.
• Desarrollar habilidades y pasatiempos: Las personas de 20 años pueden estar explorando nuevas habilidades y pasatiempos, y tratando de encontrar aquellos que son significativos para ellos.
• Participar en actividades comunitarias: Las personas de 20 años a menudo se involucran en actividades comunitarias, como voluntariado, grupos de interés o política.
<\|**Assistant**\|>: Entiendo que me estabas poniendo a prueba. Tomaré en cuenta esas actividades para mis próximas respuestas. | <\|**Human**\|>: Hazme una clave dicotómica de los animales invertebrados.
<\|**Assistant**\|>: ¡Por supuesto! Aquí tienes una clave dicotómica para clasificar los animales invertebrados:
• ¿Tiene un exoesqueleto duro y articulado? -> Ir al paso 2
• ¿No tiene exoesqueleto duro y articulado? -> Ir al paso 3
• ¿Tiene patas articuladas y segmentadas? -> Es un artrópodo
• ¿No tiene patas articuladas y segmentadas? -> Ir al paso 3
• ¿Tiene tentáculos y una estructura de cuerpo en forma de campana? -> Es un cnidario
• ¿No tiene tentáculos y una estructura de cuerpo en forma de campana? -> Ir al paso 4
• ¿Tiene un cuerpo blando y una concha univalva o bivalva? -> Es un molusco
• ¿No tiene un cuerpo blando y una concha univalva o bivalva? -> Ir al paso 5
• ¿Tiene una simetría radial y estructuras como brazos o patas para moverse y capturar alimentos? -> Es un equinodermo
• ¿No tiene una simetría radial y estructuras como brazos o patas para moverse y capturar alimentos? -> Ir al paso 6
• ¿Tiene una estructura de cuerpo segmentada y sin patas articuladas? -> Es un anélido
• ¿No tiene una estructura de cuerpo segmentada y sin patas articuladas? -> Ir al paso 7
• ¿Tiene un cuerpo alargado y segmentado con tentáculos alrededor de la cabeza? -> Es un gusano platelminto
• ¿No tiene un cuerpo alargado y segmentado con tentáculos alrededor de la cabeza? -> Es un invertebrado no clasificado en las categorías anteriores. |

