# OpenReview forum: "Improving Data Efficiency via Curating LLM-Driven Rating Systems"
_ICLR.cc/2025/Conference — ICLR 2025 Poster_

### Official Review · Reviewer_gzew · 2024-10-18

**Soundness:** 3
**Presentation:** 3
**Contribution:** 3
**Rating:** 5
**Confidence:** 4

**Summary:**

The paper presents DS2, a method for improving data efficiency in instruction tuning for LLMs. DS2 corrects inaccuracies and biases in LLM-generated data quality scores using a score transition matrix and promotes diversity in the data selection process. The authors demonstrate that using a curated subset  can outperform larger datasets, challenging traditional data scaling laws and achieving better results than human-curated datasets like LIMA. Key contributions include modeling score errors across various LLMs, developing a novel data curation pipeline, and conducting extensive experiments to validate DS2's effectiveness against multiple baselines.

**Strengths:**

1. The paper argue that a smaller, high-quality dataset can outperform larger datasets. The introduction of a score transition matrix to model LLM-based errors adds a novel approach to data curation, combining insights from label noise reduction and LLM evaluation.
2. The experimental section is substantial and persuasive, featuring a rich number of comparative experiments that cover various scenarios and conditions, effectively demonstrating the robustness and efficacy of the proposed algorithm. Furthermore, the experimental results indicate significant performance improvements, further validating the superiority of the proposed approach.

**Weaknesses:**

1. While the score transition matrix is central to DS2, the paper lacks an in-depth analysis of its limitations or conditions under which it may not be effective. For instance, the independence assumption (that the transition matrix is not sample-specific) might limit its ability to model certain data-specific errors.
2. The paper heavily relies on the capabilities of pre-trained LLMs to generate initial quality ratings and the transition matrix to correct the final quality score.
3. If there is a strong LLM evaluator (eg, GPT-4o or o1), the score error correction process seems unnecessary.

**Questions:**

1. Can the authors provide more examples of typical errors in LLM-rated scores and explain how DS2 specifically addresses these? This would clarify the practical benefits of the score transition matrix.
2. Could the authors discuss how the diversity score metric performs in datasets with different distributions or in tasks with varying levels of data complexity? Providing case studies or examples would help demonstrate the versatility of this metric.
3. How sensitive is DS2's performance to the choice of embedding model used for the k-NN agreement score? Exploring different embeddings beyond the BGE model may provide insights into the robustness of the approach.
4. In an open-ended question, the responses may not be similar at all (resulting in very different embedding vectors), but they could all be equally excellent answers. In this method, I believe the two responses may be curated into completely different scores, leading to errors. ”Intuitively, a lower cosine similarity score indicates a higher likelihood of a rating error”, This assumption does not adequately explain the examples above.
5. "please tell me the answer of below math question '1+1=?' answer:2"
"please tell me the answer of below math question '1+1=?' answer:3"
The two questions have similar embedding vectors, but they should have completely different scores. This seems to contradict the notion that samples with high embedding vector similarity are more likely to have the same labels.

---

> ### Author Response · Authors · 2024-11-21
> **Response 1/4**
>
> We want to thank the reviewer for their positive feedback and comments. We will address individual comments below.
>
>
>
> **W1: In-depth analysis of the limitations**
>
> Thank you for raising this concern.  This independence assumption is a mathematical simplification that allows us to develop our statistical approach to perform estimation work. Our assumption follows the learning with noisy label literature settings [1, 2, 3]. As it is also studied in the literature, when making no assumption of the transition matrix's dependence on the sample-level features, it becomes extremely hard to identify the correct noise structure of this noisy rating problem [4] and therefore disables the development of corresponding detection approaches. Besides, this assumption can also be viewed as the "average case" estimation of the amount of errors for samples with the same true rating (though we agree with the reviewer that it can imperfect and non-ideal)
> Empirically, our final experimental results demonstrate the efficacy of our method under this assumption. There could be a valuable future research direction to enhance performance under weaker assumptions, such as group-dependent or sample-dependent.
> We have included the limitations in the revised manuscript (Appendix A):
>
> - **Sample-independent assumption**: The sample-independent assumption is critical for deriving the transition matrix $T$ and the true score probability distribution $p$. However, this assumption may be somewhat strong and could inevitably introduce certain data-specific errors. Exploring weaker assumptions, such as group-dependent approaches, could be a valuable direction for future research.
> - **Base model scale**: Our experiments are primarily conducted on pre-trained models at the 7B/8B scale. It remains uncertain how well the method would perform on larger-scale pre-trained models.
> - **Rating models**: Due to cost considerations, we use the more affordable GPT-4o-mini to generate GPT-level scores. It is unclear whether the score curation mechanism works for more powerful GPT models (e.g., GPT-4 or GPT-o1).
> - **k-NN clusterability**. The k-NN clusterability definition implies that similar embedding vectors should correspond to the same rating score or class, a characteristic commonly leveraged in image classification tasks. However, in text-related tasks, highly similar text can convey opposite semantic meanings due to subtle differences, such as a single word change. To address this challenge, powerful embedding models are essential to accurately distinguish these subtle differences and effectively capture the underlying semantic meaning.
>
> [1] Learning with noisy labels, NeurIPS 2013.
>
> [2] Classification with noisy labels by importance reweighting, TPAMI 2015.
>
> [3] Peer loss functions: Learning from noisy labels without knowing noise rates, ICML 2020.
>
> [4] Identifiability of Label Noise Transition Matrix, ICML 2023.
>
>
> **W2: This paper heavily rely on the capabilities of pre-trained LLMs**
>
> Thank you for pointing out this concern. We would like to clarify that the characterization of this work is not a weakness. LLM-based data selection procedures have been demonstrated to be both efficient and widely adopted in practice. This is evident from several newly published works, including AlpaGasus (ICLR 2024), DEITA (ICLR 2024), Qurating (ICLR 2024), InsTag (ICLR 2023), LESS (ICML 2024), and Tree-instruct (ACL 2024), to name a few. More details regarding these methods are discussed in Section 2 (Related Work). Beyond academic research, LLM-based ratings are also extensively used in industry for data selection tasks, serving as a cost-effective alternative to manual annotations.
>
> Furthermore, even though the embeddings used are extracted from the newly released model BGE, our transition matrix does not heavily depend on its performance. In Appendix D, we investigate the impact of using SentenceBert as the embedding model. When compared to Figure 3 (BGE), the transition matrix shown in Figure 8 (SentenceBert) exhibits a striking similarity, demonstrating that our transition matrix is largely independent of the specific embedding model used.

---

> ### Author Response · Authors · 2024-11-21
> **Response 2/4**
>
> **W3: The score error correction process seems unnecessary if using a strong LLM evaluator**
>
> Thank you for pointing out this. We agree that a strong LLM evaluator could be beneficial for generating accurate ratings. However, due to the **higher cost** of using a strong LLM evaluator, current open-sourced LLMs (such as LLaMA-3.1 and Mistral) offer a cost-effective and efficient alternative. The score error correction process plays a crucial role in bridging the performance gap between GPT-based and open-sourced LLMs. Our experimental results further highlight the necessity and effectiveness of this score error correction process, particularly for GPT-4o-mini. From the experimental results of GPT-4o-mini, we believe that the phenomenon of score errors is widespread and unavoidable, even for strong LLM evaluators.
>
> Besides, we would like to clarify that **GPT may not evaluate perfectly across all tasks**. For instance, in certain domain-specific tasks, GPT may lack the necessary knowledge to provide accurate evaluations due to the privacy and restricted access to such specialized data. In such cases, GPT might produce inaccurate evaluations, leading to potential scoring errors. Therefore, implementing a score curation mechanism to mitigate potential score errors remains crucial for ensuring more reliable and accurate evaluations.
>
> **Q1: Examples of Typical Errors in LLM-Rated Scores and How DS2 Addresses Them**
>
>
> Thank you for your comment. Here, we present a target sample from our data pool along with its 2NN samples to illustrate this. By analyzing these samples manually, it becomes evident that the target sample is likely misrated and should have a score of 3.
>
> - Target Example(Score: 1): User: 'You need to complete the following task: Calculate 15\% of the following number: 100’, Assistant: '15\% of 100 is 15.'
> - KNN Example 1 (Score: 3): User: ‘Calculate 15\% of 500’. Assistant: ‘75’
> - KNN Example 2 (Score: 3): User: 'Calculate 50\% of 300.”, Assistant: ' 50\% of 300 is 150.'
>
>
>
> First, DS2 derives the score transition matrix based on the concept of $k$-NN clusterability. Specifically, it aggregates the score information from all $2$-NN clusters and calculates the average score proportion probability, which serves as the consensus information for constructing the score transition matrix [Section 3.2].
>
> The score transition matrix provides statistical transition probabilities but does not identify specific samples that are likely to be misrated. To address this, DS2 computes the $k$-NN agreement score for each sample [Section 4.1, Line 282]. Intuitively, if a sample’s $k$-NN agreement score is low—indicating its score significantly differs from the scores of its $k$-NN neighbors—it is likely to have been misrated. Samples are then ranked by their $k$-NN agreement scores, and the bottom $M$ samples are flagged as misrated, where $M$ is determined based on the error threshold derived from the statistical transition matrix. [Section 4.1, Line 291].
>
> To address imbalances in LLM-based scores, a confidence probability is introduced to adjust the size of the identified misrated sample set. Finally, DS2 curates (replaces) the scores of these misrated samples with candidate scores suggested by the $k$-NN agreement (majority vote) [Section 4.1, Line 306].
>
>  Therefore, in the above example, the score for the target sample will be curated to 3 by DS2, consistent with its 2NN neighbors.

---

> ### Author Response · Authors · 2024-11-21
> **Response 3/4**
>
> **Q2: Explore the impact of diversity score in datasets with different distributions using case studies**
>
> The importance of diversity on LLM data selection has been extensively explored by previous work [1, 2,3]. Note that our data pool is composed of five distinct subsets, each characterized by varying levels of complexity and diversity. The statistical analysis of diversity scores across subsets, as illustrated in Figure 14, confirms this. More details could be found in Appendix I of the revised version. To evaluate the versatility of diversity score, we further conduct additional contrast experiments here. In particular, we solely rank the samples of subsets based on the diversity score. Then, we select the Top-k and Bottom-k samples independently to construct datasets for LLM instruction finetuning, where k =10000 here. The corresponding performance results are presented in the following table. For cost considerations, we employ LLaMA-3.2-3B as the base model. The experimental settings are consistent with those outlined in our paper. From the table, it is evident that the diversity score is not universally effective across all datasets. To achieve better results, it should be complemented with other specific metrics, such as LLM rating scores.
>
> Table 1. Performance comparison across different subsets. Bottom-k (Top-k) refers to the samples with the lowest (highest) diversity scores, where $k$ is fixed at 10,000.
> | Model                | MMLU  | BBH   | GSM8K | TruthfulQA(MC2) | Tydiqa | Average  |
> |----------------------|-------|-------|-------|------------------|--------|----------|
> | flan_v2 (Bottom-k)         | 55.56 | 44.91 | 24.5  | 38.63           | 55.9   | 43.900   |
> | flan_v2 (Top-k)         | 54.84 | 45.00 | 29.5  | 41.69           | 60.5   | 46.306   |
> | wizardlm(Bottom-k)        | 56.68 | 45.83 | 30.5  | 46.64           | 37.7   | 43.470   |
> | wizardlm(Top-k)        | 56.60 | 47.69 | 28.5  | 48.15           | 31.2   | 42.428   |
> | Alpaca (Bottom-k)  | 56.50 | 46.30 | 28.5  | 40.19           | 48.4   | 43.978   |
> | Alpaca(Top-k) | 55.13 | 47.13 | 26.0  | 40.58           | 39.5   | 41.668   |
>
> [1] How Far Can Camels Go? Exploring the State of Instruction Tuning on Open Resources, NeurIPS 2023.
>
> [2] What makes good data for alignment? a comprehensive study of automatic data selection in instruction tuning, ICLR 2024.
>
> [3] Self-Instruct: Aligning Language Models with Self-Generated Instructions, ACL 2023.
>
> **Q3: Explore the impact of embedding models**
>
> Thank you for your insightful comment. In the Appendix D (line 1036), we utilize the typical SentenceBERT Model to extract the embeddings of data samples. The corresponding score transition matrix as well as the score pattern shown in Figure 8  is similar to the matrix using the BGE model (Figure 3). Therefore, one reasonably claim that the impact of embedding space is limited, the choice of embedding model does not significantly affect the error patterns produced by LLMs, which also highlight the robustness of our approach.
>
>
> **Q4: Concerns Regarding Cosine Similarity Assumptions and Potential Errors in Open-Ended Questions**
>
> We would like to clarify that the k-NN agreement score is defined as the average cosine similarity LLM score distance among k-NN samples. The intuition is that when samples with similar embedding vectors have significantly different scores, resulting in a lower k-NN agreement score, the uncertainty of the LLM rating for this example increases. Consequently, such samples are more likely to require curation. Whether a sample needs curation and whether a curated score should be assigned finally is determined jointly by an error threshold and confidence probability.
>
> When the embedding vectors of two responses to the same open-ended question differ significantly, these two samples can be considered independent, as their k-NN samples may not include each other. In such cases, their cosine similarity scores are calculated based on their respective k-NN samples independently.

---

> ### Author Response · Authors · 2024-11-21
> **Response 4/4**
>
> **Q5: Contradiction in Embedding Similarity and Label Consistency for Math Questions**
>
>
> Thank you for bringing up this important point! We partially agree with the claim that these two questions might conflict with our notion. The scores of samples are influenced not just by correctness but also by broader quality metrics, such as rarity, complexity, and informativeness, as emphasized in our prompt template.
> Focusing solely on correctness with binary scores (0 or 1) treats correct and incorrect answers distinctly. However, when assessing overall quality on a granular scale (e.g., [0–10], later compressed to [0–5]), both questions could score 0 for simplicity and low informativeness. Thus, considering overall quality reduces the direct emphasis on correctness while reinforcing the evaluation framework. To evaluate this, we generate the scores from the above two questions (Example 1 \& Example 2) in the same paper setting.
>
> - **Example 1**: please tell me the answer of below math question '1+1=?' answer:2"
> - **Example 2** "please tell me the answer of below math question '1+1=?' answer:3"
> - **Example 3** "please tell me the answer of below math question '10+10=?' answer:20"
> - **Example 4** “Calculate 15% of 500.\n\n answer: 75”
> - **Example 5** “Calculate 50% of 300. answer: '50% of 300 is 150.”
>
> Table 2. LLM rating scores comparison between Example 1 and Example 2.
>
> | Metric             | LLaMA Example1 | LLaMA Example2 | Mistral  Example1 | Mistral  Example2 | Gemma  Example1 | Gemma  Example2 | GPT  Example1 | GPT Example2 |
> |--------------------|-----------------|-----------------|-------------------|-------------------|-----------------|-----------------|------------------------|------------------------|
> | Rarity            | 2               | 1               | 2                 | 1                 | 3               | 2               | 2                      | 1                      |
> | Complexity        | 1               | 1               | 1                 | 1                 | 1               | 1               | 1                      | 1                      |
> | Informativeness   | 1               | 4               | 5                 | 4                 | 3               | 3               | 1                      | 2                      |
> | Overall Rating    | 3               | 3               | 4                 | 3                 | 4               | 3               | 2                      | 1                      |
>
>
> Furthermore, in practice, the embedding models are able to capture the semantic similarities and dissimilarities instead of accumulating the similarities at the token level.
> We generate the embedding vectors of several examples using four embedding models, including GPTs. Here, Examples 4 and 5 are taken from the data pool, while Example 3 is a revised version of Example 1. We observe that even small differences result in much larger embedding cosine similarity distances, effectively capturing the additional semantic information.
>
> Table 3. Cosine similarity distances between any two examples.
> |      Embedding Model            | Example 1 & 2 | Example 2 & 3 | Example 4 & 5 |
> |-------------------------|---------------|---------------|---------------|
> | GPT text-embedding-3-small  | 0.0208        | 0.1380        | 0.1415        |
> | GPT text-embedding-3-large  | 0.0490        | 0.2043        | 0.2464        |
> | GPT text-embedding-ada-002  | 0.0050        | 0.0487        | 0.0697        |
> |bge-large-en-v1.5 |  0.00772 | 0.04975 | 0.09283 |
>
> In our revised manuscript, we include some target samples with their 2-NN examples from our data pool in **Table 10** (Page 21) to evaluate k-NN clusterability. For simplicity of illustration, we filter out too long samples. From these randomly selected examples, we find that our concept holds true for the data pool, and these extreme questions are generally unlikely to occur. For this point, the issue is significantly mitigated. Besides, from our observation, the data samples from our data pool (Flan_v2, WizardLM, Oasst1, Alpaca and Dolly) are almost correct but vary in quality level. Therefore, correctness is not our main goal (we fail to evaluate the data sample using the correctness), and then the above problem resulting from the correctness problem should be fine in our paper.

---

> > ### Author Response · Authors · 2024-11-26
> > **Follow-up response about the k-NN clusterability hypothesis (Q5)**
> >
> > Based on Reviewer XKFp’s recommendation, we have conducted a more systematic analysis instead of several examples to further validate and demonstrate the practicality of k-NN clusterability, as outlined below.
> >
> >  **Systematic analysis about the k-NN clusterability definition** Due to the unavailability of samples’ ground truth scores, we evaluate k-NN clusterability by examining the distribution of **average score gaps**, which measures the score difference within one k-NN cluster.
> > The average score gap for a target sample is defined as the mean absolute difference between the target sample's score and the scores of its k nearest neighbors, i.e., $\texttt{average score gap} = \texttt{Mean}(|\texttt{target sample’s score - kNN sample’s score}|)$. In our work, we focus on **2-NN clusterability** and frame our analysis within this context. Specifically, for each 2-NN cluster, we consider a target sample and its two nearest neighbors.  For example, given a 2-NN cluster with the score tuple: (target sample: 1, kNN sample 1: 2, kNN sample 2: 3), the score gap is calculated as:
> > $
> > \text{Average score gap} = \frac{|1 - 2| + |1 - 3|}{2} = 1.5.
> > $
> > The following Table 1 below summarizes the statistical distribution of score gaps across all 2-NN clusters.
> >
> > Table 1. Average score gap statistical information of all 2-NN clusters from our data pool.
> > | Curation        | Model                        | Score Gap (0.0-1.0) (%) | Score Gap 1.5 (%) | Score Gap 2.0 (%) | Score Gap >2.0 (%) |
> > |------------------|------------------------------|----------------------|-------------------|-------------------|---------------------|
> > | w/o Curation    | GPT                 | 81.0                 | 12.0              | 4.9               | 2.1                 |
> > | w/o Curation    | LLaMA  | 58.3                 | 18.0              | 12.2              | 11.5                |
> > | w/o Curation    | Mistral    | 70.2                 | 16.5              | 8.1               | 5.4                 |
> > | w/ Curation     | GPT                 | 82.5                 | 10.9              | 4.5               | 1.7                 |
> > | w/ Curation     | LLaMA  | 78.8                 | 9.4               | 7.3               | 4.1                 |
> > | w/ Curation     | Mistral    | 80.5                 | 10.8              | 5.6               | 4.3                 |
> >
> >
> >
> >
> > From Table 1, we observe that **without score curation**, GPT has a higher proportion of samples in the 0.0–1.0 score gap range (81.0%) compared to Mistral (70.2%) and LLaMA (58.3%). This reveals that more powerful rating models, such as GPT, tend to exhibit smaller average score gaps, which aligns more closely with the concept of **k-NN clusterability** and contributes to improved performance.
> >
> >
> > Moreover, when comparing the settings **with and without score curation**, we observe that all three rating models show an increased proportion of samples in the 0.0–1.0 score gap range after score curation. From Table 2, this shift in the score gap distribution correlates strongly with the performance improvements observed on LLM leaderboard tasks (as detailed in Table 3 of the manuscript).
> >
> >
> > Table 2. The proportion of samples in the 0.0–1.0 score gap range both with and without score curation for each rating model. For comparison, the corresponding average performance on LLM Leaderboard tasks is included in parentheses.
> >
> >
> > | Rating Model | 0.0-1.0 score gap w/o Curation (Average Performance) | 0.0-1.0 score gap w/ Curation (Average Performance)|
> > |--------------|------------------|---------------|
> > | GPT          | 81.0% (60.2)          | 82.5%   (61.4)      |
> > | LLaMA        | 58.3% (59.2)          | 78.8% (60.2)        |
> > | Mistral      | 70.2%  (60.7)         | 80.5%  (61.1)       |
> >
> >
> > Therefore, these results demonstrate the validity of the **k-NN clusterability** definition proposed in our paper. For a clearer visualization of score gap proportions before and after score curation, we encourage the reviewer to refer to the newly added **Figure 8**. For your reference, the average embedding distance distribution across all 2-NN clusters from our data pool is also provided in the newly added **Figure 9** in the revised manuscript.

---

### Official Review · Reviewer_XKFp · 2024-11-03

**Soundness:** 3
**Presentation:** 3
**Contribution:** 2
**Rating:** 6
**Confidence:** 3

**Summary:**

The paper proposed the data selection pipeline named “DS^2”, which utilizes the kNN statistical information based on the rating scores to curate ratings, and then achieve better data selection. The authors validated their methods by conducting experiments with various rating models and benchmark tasks.

**Strengths:**

Originality: The paper proposed a novel method to utilize the score transition matrix to detect scoring errors. Although the score transition matrix is proposed by previous works, the application of that into LLM data selection is novel.

Quality: The paper provides detailed explanations of their methods and validates the results using thorough experiments, in which they include three models (GPT-4o-mini, Llama-3.1-8B-Instruct, Mistral-7B-Instruct-v0.3) and various benchmarking tasks (MMLU, TruthfulQA, etc). Moreover, they also evaluated the alignment performance. All these greatly improve the quality of the paper.

Clarity: The paper demonstrated great clarity when exhibiting the error patterns of LLM scores, as well as when explaining the experiment setup and results.

Significance: The significance is validated with the experiment results and the ablation studies.

**Weaknesses:**

The paper does not include discussions about the efficiency of the proposed method and the efficiency comparison with other baseline methods. Part of the computation relies on estimating the score transition matrix, but the details about this estimation are lacking. Although the score transition matrix is proposed and discussed in previous works, it is still necessary to analyze their cost within the current framework.  Moreover, the paper does not include any baselines that also improve data quality by modifying the score ratings of the data. The current framework essentially proposed a method to modify the rating scores and then conduct data selection based on new scores. It would be helpful to further validate the method by comparing other methods that also study LLM ratings.

**Questions:**

I find the discussion around line 200, saying that similar embeddings should belong to the same category, not very convincing. Because here each class is the rating score, in my opinion, two samples could have similar textual embeddings but completely different correctness or quality, which lead to distinct rating scores. I think it is crucial to better explain this intuition behind the proposed method.

---

> ### Author Response · Authors · 2024-11-21
> **Response 1/2**
>
> We want to thank the reviewer for their positive feedback and comments. We will address individual comments below.
>
>
>
> **W1: The efficiency of the proposed method and the efficiency comparison with other baseline methods**
>
> Thank you for your insightful suggestion. We have summarized the storage and runtime of our approach alongside three baselines below. The wall-clock time was measured on a Microsoft Azure cluster with 8 A100 (80GB) GPUs. Note that our score curation mechanism relies primarily on linear programming (LP), which runs exclusively on the CPU. As shown in the table, LLM rating systems are advantageous over the gradient-based method LESS in terms of both storage and runtime. Notably, compared to AlpaGasus and DEITA, our method avoids any significant computation costs on the GPU.
>
> Table 1. Comparison of storage and running time over baselines.
>
> | Method     | Storage  |  Running Time ||  |                                | Base Model Free | Validation Set |
> |------------|----------|------------------|---|----|----------------------|-----------------|----------------|
> |            |          | Rating/Gradient | Diversity Score | CPU-only Curation | Data Selection |                 |                |
> | LESS       | 20GB     | 66 hours            | -               | -                 | <1min          | No              | Required       |
> | AlpaGasus  | <10MB    | 6 hours            | -               | -                 | <1min          | Yes             | Not Required   |
> | DEITA      | <10MB    | 6 hours             | 10 mins         | -                 | <1min          | Yes             | Not Required   |
> | Ours       | <10MB    | 6 hours             | -               | 15 mins           | <1min          | Yes             | Not Required   |
>
> **W2: Baselines by modifying the score ratings**
>
> Thank you for your thoughtful suggestion. We would like to clarify that to our best knowledge, our work is the first one to incorporate the score curation mechanism into the rating-based LLM data selection procedures. Although the development of LLMs has been progressing rapidly, the application of LLM ratings remains a relatively new area of research. Starting from AlpaGasus [1] (ICLR 2024),  existing works that study LLM ratings both directly apply the raw scores generated from LLMs to do data selection. DEITA [2] (ICLR 2024) is a follow-up work by jointly considering data diversity. We will really appreciate it if the reviewer could provide any reference about modifying the score ratings.
>
> [1] AlpaGasus: Training a Better Alpaca with Fewer Data, ICLR 2024.
>
> [2] What Makes Good Data for Alignment? A Comprehensive Study of Automatic Data Selection in Instruction Tuning, ICLR 2024.

---

> ### Author Response · Authors · 2024-11-21
> **Response 2/2**
>
> **Q1: Concerns about the KNN clusterability definition that similar embeddings should belong to the same category**
>
> Thank you for your insightful comment!
> We agree with the reviewer that the proposed method depends on the quality of embeddings. In some tasks, if we use a general embedding model, e.g., using a general textural embedding model for math problems, may not be the best choice since changing a single number can easily destroy the correctness of the answer, thereby resulting in a different  LLM rating score. And the correctness would be the most important factor in changing LLM rating scores of samples.  This warns us to get or use an appropriate embedding model for specific tasks, i.e., a domain-specific embedding model [1].
> The statement (a.k.a, k-NN clusterability definition) is based on the assumption that embeddings capture semantic and contextual similarity for textual data, which often correlates with quality and correctness. Similar to image classification tasks, these high-dimensional representations map semantically similar texts to nearby points in the vector space while positioning dissimilar texts farther apart, enabling clustering that aligns with classification categories. We acknowledge that samples with subtle token-level differences can yield different scores due to variations in correctness (the most important factor leading to distinct rating scores), our scoring approach considers not just correctness but also overall quality metrics such as rarity and informativeness, as outlined in our prompt template. This helps mitigate the influence of correctness alone on the final score. To evaluate this, we generate the scores from the below two simple but extreme examples (Example 1 and Example 2) in the same paper setting.
> - **Example 1**: please tell me the answer of below math question '1+1=?' answer:2"
> - **Example 2** "please tell me the answer of below math question '1+1=?' answer:3"
> - **Example 3** "please tell me the answer of below math question '10+10=?' answer:20"
> - **Example 4** “Calculate 15% of 500.\n\n answer: 75”
> - **Example 5** “Calculate 50% of 300. answer: '50% of 300 is 150.”
>
> Table 1. LLM rating scores comparison between Example 1 and Example 2.
> | Metric             | LLaMA Example1 | LLaMA Example2 | Mistral Example1 | Mistral Example2 | Gemma Example1 | Gemma Example2 | GPT Example1 | GPT Example2 |
> |--------------------|-----------------|-----------------|-------------------|-------------------|-----------------|-----------------|------------------------|------------------------|
> | Rarity            | 2               | 1               | 2       | 1        | 3         | 2       | 2             | 1             |
> | Complexity        | 1               | 1               | 1                 | 1                 | 1               | 1               | 1              | 1                      |
> | Informativeness   | 1               | 4               | 5                 | 4                 | 3               | 3               | 1                      | 2                      |
> | Overall Rating    | 3               | 3               | 4                 | 3                 | 4               | 3               | 2                      | 1                      |
>
> Furthermore, in practice, using a powerful embedding model is essential for capturing the semantic information under subtle token-level differences. We estimate the embedding vectors using several embedding models, including GPTs. Here, Examples 4 and 5 are taken from the data pool, while Example 3 is a revised version of Example 1. We observe that even small differences result in much larger embedding cosine similarity distances, effectively capturing the additional semantic information.
>
> Table 2. Cosine similarity distances between any two examples.
> |   Embedding   Model  | Example 1 & 2 | Example 2 & 3 | Example 4 & 5 |
> |-------------------------|---------------|---------------|---------------|
> | GPT text-embedding-3-small | 0.0208| 0.1380 | 0.1415|
> | GPT text-embedding-3-large | 0.0490 | 0.2043| 0.2464|
> | GPT text-embedding-ada-002 | 0.0050 | 0.0487 | 0.0697|
> |bge-large-en-v1.5 | 0.00772| 0.04975 | 0.09283 |
>
> In our revised manuscript, we include some 2-NN examples from our data pool in **Table 10** (Page 21) to evaluate k-NN clusterability. From these randomly selected examples, we find that our concept holds true for the data pool, and these extreme questions are generally unlikely to occur. For this point, the issue is significantly mitigated. Besides, from our observation, the data samples from our data pool (Flan_v2, WizardLM, Oasst1, Alpaca and Dolly) are almost correct but vary in quality level. Therefore, correctness is not our main goal (we fail to evaluate the data sample using the correctness), and then the above problem resulting from the correctness problem should be fine in our paper.
>
> [1] Do We Need Domain-Specific Embedding Models? An Empirical Investigation, arxiv 2024.

---

> ### Comment · Reviewer_XKFp · 2024-11-24
>
> Thanks to the authors for the responses. To address my concern, more systematic analysis might be needed instead of a few extreme examples.  Thus the responses address my concerns only to a very limited extent. I will maintain my original score.

---

> > ### Author Response · Authors · 2024-11-25
> > **More systematic analysis about k-NN clusterability definition**
> >
> > Dear Reviewer XFFp,
> >
> > It is great to hear that some of your concerns have been addressed! Thank you for your thoughtful  follow-up comments! We would like to provide more systematic analysis here to address your concerns.
> >
> >  **Systematic analysis about the k-NN clusterability definition** Due to the unavailability of samples’ ground truth scores, we evaluate k-NN clusterability by examining the distribution of **average score gaps**, which measures the score difference within one k-NN cluster.
> > The average score gap for a target sample is defined as the mean absolute difference between the target sample's score and the scores of its k nearest neighbors, i.e., $\texttt{average score gap} = \texttt{Mean}(|\texttt{target sample’s score - kNN sample’s score}|)$. In our work, we focus on **2-NN clusterability** and frame our analysis within this context. Specifically, for each 2-NN cluster, we consider a target sample and its two nearest neighbors.  For example, given a 2-NN cluster with the score tuple: (target sample: 1, kNN sample 1: 2, kNN sample 2: 3), the score gap is calculated as:
> > $
> > \text{Average score gap} = \frac{|1 - 2| + |1 - 3|}{2} = 1.5.
> > $
> > The following Table 1 below summarizes the statistical distribution of score gaps across all 2-NN clusters.
> >
> > Table 1. Average score gap statistical information of all 2-NN clusters from our data pool.
> > | Curation        | Model                        | Score Gap (0.0-1.0) (%) | Score Gap 1.5 (%) | Score Gap 2.0 (%) | Score Gap >2.0 (%) |
> > |------------------|------------------------------|----------------------|-------------------|-------------------|---------------------|
> > | w/o Curation    | GPT                 | 81.0                 | 12.0              | 4.9               | 2.1                 |
> > | w/o Curation    | LLaMA  | 58.3                 | 18.0              | 12.2              | 11.5                |
> > | w/o Curation    | Mistral    | 70.2                 | 16.5              | 8.1               | 5.4                 |
> > | w/ Curation     | GPT                 | 82.5                 | 10.9              | 4.5               | 1.7                 |
> > | w/ Curation     | LLaMA  | 78.8                 | 9.4               | 7.3               | 4.1                 |
> > | w/ Curation     | Mistral    | 80.5                 | 10.8              | 5.6               | 4.3                 |
> >
> >
> >
> >
> > From Table 1, we observe that **without score curation**, GPT has a higher proportion of samples in the 0.0–1.0 score gap range (81.0%) compared to Mistral (70.2%) and LLaMA (58.3%). This reveals that more powerful rating models, such as GPT, tend to exhibit smaller average score gaps, which aligns more closely with the concept of **k-NN clusterability** and contributes to improved performance.
> >
> >
> > Moreover, when comparing the settings **with and without score curation**, we observe that all three rating models show an increased proportion of samples in the 0.0–1.0 score gap range after score curation. From Table 2, this shift in the score gap distribution correlates strongly with the performance improvements observed on LLM leaderboard tasks (as detailed in Table 3 of the manuscript).
> >
> >
> > Table 2. The proportion of samples in the 0.0–1.0 score gap range both with and without score curation for each rating model. For comparison, the corresponding average performance on LLM Leaderboard tasks is included in parentheses.
> >
> >
> > | Rating Model | 0.0-1.0 score gap w/o Curation (Average Performance) | 0.0-1.0 score gap w/ Curation (Average Performance)|
> > |--------------|------------------|---------------|
> > | GPT          | 81.0% (60.2)          | 82.5%   (61.4)      |
> > | LLaMA        | 58.3% (59.2)          | 78.8% (60.2)        |
> > | Mistral      | 70.2%  (60.7)         | 80.5%  (61.1)       |
> >
> >
> > Therefore, these results demonstrate the validity of the **k-NN clusterability** definition proposed in our paper. For a clearer visualization of score gap proportions before and after score curation, we encourage the reviewer to refer to the newly added **Figure 8**. For your reference, the average embedding distance distribution across all 2-NN clusters from our data pool is also provided in the newly added **Figure 9** in the revised manuscript.
> >
> > Finally, we would really appreciate it if Reviewer XFFp can provide any suggestions about the systematic analysis.

---

> ### Author Response · Authors · 2024-12-03
> **Looking forward to following up**
>
> Dear Reviewer XKFp,
>
> We have revised the paper and added many additional systematic analysis about the kNN clusterability hypothesis to address your comments. Since the rebuttal period is closing very soon, can you please check the response to see whether it mitigates your concerns? We would greatly appreciate that!
>
> Thank you,
>
> The authors

---

### Official Review · Reviewer_81Qt · 2024-11-04

**Soundness:** 3
**Presentation:** 2
**Contribution:** 2
**Rating:** 6
**Confidence:** 4

**Summary:**

This study tackles the problem of rating samples using LLM, more specifically, how to improve the accuracy of LLM-based rating methods. To this end, DS2 (Diversity-Aware Score Curation for Data Selection) is proposed to model rating error pattens using a score transition matrix. Experiments on machine-alignment benchmarks show that strong results can be achieved by using just 3.3% of the original dataset.

**Strengths:**

* The idea of detecting error pattern is novel. Its implementation using score transition matrix was validated experimentally.

**Weaknesses:**

* The description of "deriving the score transition matrix" can be improved. Not sure I can reproduce it based on the given explanation. Some examples can help, which can be included in appendix.
* It is assumed that "the transition matrix is independent of sample-level features". This can be a limitation as each LLM have its own strengths and weakness (i.e., better/worse at some samples).
* Sometimes, the improvement is small, within the variance of different runs.
* Some discussions about limitations would be appreciated.

**Questions:**

* It looks like that there is an advantage of using GPT-40 mini. How to obtain embeddings from proprietary LLMs?
* Figure 1 reads like ratings of multiple LLMs are combined. The experimental results say otherwise.
* The proposed approach sometimes perform much worse (e.g., in Table 2). Is it possible to decide the appropriate conditions for using the proposed approach (and which LLM)?

---

> ### Author Response · Authors · 2024-11-21
> **Response 1/2**
>
> We would like to thank Reviewer 81Qt for the time and effort invested in reviewing this work. We will address individual comments below.
>
> **W1: Improving the Explanation of Score Transition Matrix Derivation with Examples**
>
> Thank you for your insight comment! In Appendix C.1 of the revised manuscript, we add one intuitive binary example to describe the derivation of the score transition matrix in detail. In particular, we introduce more descriptions about the consensus equations. For more details, we refer the reviewer to Appendix C.1 (Lines 858-900).  For reproduction, we will release the code as well as the generated dataset.
>
> **W2: Concerns about the impact of sample-level independent assumption for score transition matrix**
>
> Thank you for raising this insightful concern. We acknowledge that the assumption could be a limitation of this work and may potentially lead to data-specific errors. However, we would like to clarify that this assumption is a mathematical simplification that allows us to develop our statistical approach to perform estimation work. Our assumption follows the learning with noisy label literature settings [1, 2, 3]. As it is also studied in the literature, when making no assumption of the transition matrix's dependence on the sample-level features, it becomes extremely hard to identify the correct noise structure of this noisy rating problem [4] and therefore disables the development of corresponding detection approaches.
>
>  Besides, this assumption can also be viewed as the "average case" estimation of the amount of errors for samples with the same true rating (though we agree with the reviewer that it can imperfect and non-ideal).
> Empirically, our final experimental results demonstrate the efficacy of our method under this assumption. There could be a valuable future research direction to enhance performance under weaker assumptions, such as group-dependent or sample-dependent.  In Appendix A of the revised manuscript, we highlight some potential limitations including the sample-level independent assumption.
>
>
> [1] Learning with noisy labels, NeurIPS 2013.
>
> [2] Classification with noisy labels by importance reweighting, TPAMI 2015.
>
> [3] Peer loss functions: Learning from noisy labels without knowing noise rates, ICML 2020.
>
> [4] Identifiability of Label Noise Transition Matrix, ICML 2023.
>
>
> **W3: Limited performance improvement**
>
> We would like to clarify that in Figure 7, we analyze the data scaling effects of baselines across various rating models under three random seeds. The results clearly show that our method consistently outperforms the baselines. Moreover, the improvements are significantly larger than the variance observed across different runs.
>
> **W4: Discussions about the limitations**
>
> Thank you for raising this concern. While our proposed method demonstrates competitive performance compared to other baselines, we acknowledge that there are still potential limitations. We have included these limitations in the revised manuscript (Appendix A):
>
> - **Sample-level independent assumption**: The sample-independent assumption is critical for deriving the transition matrix $T$ and the true score probability distribution $p$. However, this assumption may be somewhat strong and could inevitably introduce certain data-specific errors. Exploring weaker assumptions, such as group-dependent approaches, could be a valuable direction for future research.
> - **Base model scale**: Our experiments are primarily conducted on pre-trained models at the 7B/8B scale. It remains uncertain how well the method would perform on larger-scale pre-trained models.
> - **Rating models**: Due to cost considerations, we use the more affordable GPT-4o-mini to generate GPT-level scores. It is unclear whether the score curation mechanism works for more powerful GPT models (e.g., GPT-4 or GPT-o1).
>
> For your interest, we add more discussion about the K-NN clusterability definition in Appendix A.
>
>
> **Q1: How to obtain embeddings from proprietary LLMs**
>
>  In practice, leveraging more powerful LLMs (e.g., GPT-4o-mini) can offer the advantage of reducing score errors. In our work, we consistently use a newly open-sourced BGE embedding model (as mentioned in line 173) to extract data sample embeddings for implementing the score curation mechanism. Besides, in the Appendix, we also utilize SentenceBert as the embedding model to explore the impact of embedding models, as shown in Figure 8. We observe that the impact of embedding space is limited, the choice of embedding model does not significantly affect the error patterns produced by LLMs.

---

> ### Author Response · Authors · 2024-11-21
> **Response 2/2**
>
> **Q2: Concerns about Figure 1**
>
> Thank you for bringing this to our attention. To clarify, in our paper, the three LLMs are used to independently generate raw scores for data samples, as reflected in the experimental results. To avoid misunderstanding, we have updated Figure 1 in the revised version. Additionally, in Appendix G.5 (Table 17, Line 1491), out of interest, we explore a new baseline, where ratings from multiple LLMs are combined.
>
> **Q3: Concerns about the performance of our proposed method**
>
> Thank you for your comment. For clarification, in the revised manuscript, we highlight the best results in boldface and the second-best results with underlining for different settings in our main results (Table 3). From this updated Table 3, it is evident that, in most cases, our proposed method achieves either the best or the second-best performance across various evaluation tasks. Notably, for the MMLU dataset, which emphasizes factual knowledge, prior studies have already underscored the difficulty of achieving significant performance improvements [1, 2, 3].  Thus, it is reasonable to expect that the performance gains on the MMLU task may be relatively modest.
>
> [1] Measuring Massive Multitask Language Understanding, ICLR 2021.
>
> [2] GPT-4 Technical Report, arxiv 2023.
>
> [3] Language Models are Few-Shot Learners, arxiv 2020.

---

### Official Review · Reviewer_ZhKe · 2024-11-05

**Soundness:** 3
**Presentation:** 2
**Contribution:** 3
**Rating:** 6
**Confidence:** 4

**Summary:**

The authors introduce a data curation algorithm, DS^2, to correct inaccuracies in LLM-based data quality evaluation strategies.  The authors first show how, leveraging recent work in noisy label estimation given k-NN clusterability, both the Transition matrix and true probability distribution of LLM scores may be estimated by solving a linear program.  Under this k-NN clusterability assumption, the authors demonstrate that widely used LLM scorers (GPT-4o-mini, Llama-3.1 8B, and Mistral 7B) provide incorrect scores.  Using both Transition matrix and (true score) probability distribution, the layout the steps of their DS^2 algorithm and calculation of critical quantities: the error threshold for classifying samples as misrated, the confidence probability for mitigating imbalances in LLM scores, and the long-tail diversity score.

The authors then compare several data selection experiments leveraging the DS^2 algorithm over a large data pool (300k) containing several widely used datasets (i.e., Alpaca, Dolly, Flan V2, etc.).  Compared to several data selection heuristics and competitor LLM-based data selection algorithms (i.e., AlpaGasus and Deita), the authors show that their methods outperforms competitors per-LLM-judge and averaged over common natural language benchmarks (MMLU, TruthfulQA, GSM, BBH, and TydiQA) at a data selection size of 10k samples.  Representative experiments than compare the performance of DS^2 selecting 1k samples (over the 300k data pool) compared to the LIMA dataset (which was heavily vetted and selected using human feedback), once again outperforms finetuning LLama-3.1-8B and Mistral 7B models using the LIMA dataset.  Scaling law experiments as well as hybrid data selection schemes (e.g., AlpaGasus leveraging DS^2's data curation scores) further show the efficacy of the presented methods.

# Post-rebuttal update
The authors have posted additional experiments to address my major concerns.  Particularly, an apples to apples comparison to AlpaGasus demonstrating DS^2's superiority and a discussion/experiments tackling the practical aspects of k-NN clusterability for this use case.  I am raising my score from 5 to 6.

**Strengths:**

The procedure to estimate the transition matrix from LLM scores is intuitive.  The beginning of the paper is also well written, and the examples showing the deficiencies in LLM-based scores is compelling.  The subsequent probability quantities leveraged throughout the DS^2 algorithm also largely make sense.  The large number of experiments demonstrating the effectiveness of the approach is also compelling.

**Weaknesses:**

Firstly, I would like to congratulate the authors on the DS^2 algorithm and the idea of leveraging k-NN clusterability to correct LLM-based scores.  Such a line of work seems promising.  With that said, I have several concerns that I am hoping can be addressed (I am greatly looking forward to the authors feedback).

While the beginning of the paper is well written, there seemed to be either missing details or a lack clarity for important concepts.  In particular, Algorithm 1 is difficult to follow; "CuratedScores" is not defined in Algorithm 1. Are "ConfidenceProbs" the curated scores?  Furthermore:
> the average likelihood of identifying the sample n as misrated over multiple epochs.

How is this calculated in practice?  Given a dataset with an arbitrary ordering of samples, how can a statistic dealing with the arbitrary index $n$ be meaningful?  Put another way, this calculates the statistics at the sample index, but the data pool samples (line 172) are not stated as being ordered in any specific way.  Thus, how can a statistic calculated over the sample index be anything other than arbitrary?

Furthermore, while the notion of k-NN score clusterability (as defined in Definition 3.2) is assumed, this notion is never actually demonstrated for the LLM judges and datasets considered.  Do the score distributions depicted in FIgure 2 abide by k-NN score clusterability?  If not, could the authors speak to how this affects the Transition matrix and (true score) probability distribution calculations from the concensus vectors (Equation 1)?  If k-NN score clusterability is violated in practice, it does not seem possible to guarantee the correctness of estimating the Transition matrix and (true score) probability distribution in this case.

There were some concerns regarding the evaluations of AlpaGesus:
> AlpaGasus (Chen et al., 2023) utilizes ChatGPT to rate
data samples and solely select high-rated samples"

One of the major innovations of AlpaGasus was the prompt template.  Was this used for the AlpaGasus results in Table 3?  Note that they also showed the training data size is a very important hyperparameter for Alpaca; thus, a fairer/more apples-to-apples comparison would have been DS^2 applied to Alpaca with a budget of 9k samples and compared to fine-tuning performance (Table 3) of the AlpaGasus dataset.

**Questions:**

For the k-NN agreement score, assuming the k-NN clusterability characteristic, isn't the similarity score just always unity?  I.e., due to clusterability, the frequency will only occur in the same index as the one-hot encoding vector, and thus the numerator and denominator turn out to be the same.  Also, have the authors empirically explored whether using the embedding feature-vector or the one-hot encoding rated score vector work better in practice?

On the initial read through the paper, there was some initial confusion between: the avoidance of relying on the ground truth scores for DS^2, yet the appearance of the ground truth per-class probability in Section 4.  However, after careful rereading, we have the following: assuming k-NN score clusterability, we are thus able to leverage the concensus vectors and solve the LP for \mathbf{T} and \mathbf{p} (I understand this is stated, but it is easy to miss this simple statement buried in lines 231-241).  I would recommend either reiterating that solving the LP formed from Equation 1 provides both the transition matrix and ground truth probability vector, or breaking up the paragraph on lines 231-241 and emphasizing this.

"the average likelihood of identifying the sample n as misrated over multiple epochs." <- How is this calculated in practice?

For Completion Length, please cite the following:
Zhao, Hao, et al. "Long is more for alignment: A simple but tough-to-beat baseline for instruction fine-tuning." arXiv preprint arXiv:2402.04833 (2024).

For Table 3, please specify the data pool is listed in Table 2, e.g., " Performance comparison on OpenLLM leaderboard, using the data pool listed in Table 2."

Could the authors please bold the winners of each column (per grouping) in Table 3?

---

> ### Author Response · Authors · 2024-11-21
> **Response 1/3**
>
> We want to thank reviewer ZhKe  for the positive feedback and comments. We will address individual comments below.
>
>
> **W1: missing algorithmic details and the lack clarify for important concepts**
>
> Thank you for highlighting this. To clarify, "confidenceProbs" are not the curated scores used in this paper. The "CuratedScores" are those raw rating scores which may be curated by our score curation mechanism. These confidence probes are only used to identify misrated samples, where raw scores were then replaced by the majority of KNN scores. We realized that one line of code was missing at the end of the score curation procedure, shown as $ \texttt{CuratedScores} = \texttt{ScoreCuration}(\texttt{MisratedSamples}, \texttt{ConfidenceProbs})$.
> Besides, in Appendix C of the revised manuscript, we add one intuitive binary example to describe the derivation of the score transition matrix in detail.
>
> **W2: the practical calculation of $\overline{p}_n$**
>
> Thank you for pointing this out. There seems to be a misunderstanding. In Line 314, $\overline{p}_n$, which represents the average likelihood of identifying sample $n$ as misrated over multiple epochs, is used to calculate the confidence probability. We would like to clarify that the sample index $n$ is used purely for identification purposes and not for ordering. This means that the value of $\overline{p}_n$ for individual samples remains unchanged regardless of the calculation sequence between any two examples, such as (5th example, 6th example) or (6th example, 5th example). In practice, we determine whether a sample is misrated over multiple cleaning epochs by assigning a value of 1 if the sample is identified as misrated and 0 otherwise.  Specifically, the value of $\overline{p}_n$ is calculated as the average of the misrated labels (values of 0 or 1) across multiple epochs, effectively reflecting the proportion of times the sample is identified as misrated, thereby making the detection process more robust against noise. For example, suppose the misrated labels of one example over five epochs are {0, 1, 0, 1, 1}. Then $\overline{p}_n$ will be $0.6$.
>
>
> **W3: Concerns on k-NN score clusterability and its impact on transition tatrix estimation**
>
> We would like to clarify that the score distribution in Figure 2 solely represents the statistical characteristics of rating scores and is not influenced by the embeddings of the samples. As a result, the score distribution alone cannot capture k-NN score clusterability, which is inherently linked to the samples’ embeddings.
> In fact, the k-NN clusterability ensures that k-NN information is meaningful and captures the relationship between embedding vectors and quality rating scores, such as potential score errors or score transition probabilities.
>
> - **Target Example (Score: 1)**: User: 'You need to complete the following task: Calculate 15\% of the following number: 100’, Assistant: '15\% of 100 is 15.'
> - **KNN Example 1 (Score: 3)**: User: ‘Calculate 15\% of 500’. Assistant: ‘75’
> - **KNN Example 2 (Score: 3)**: User: 'Calculate 50\% of 300.”, Assistant: ' 50\% of 300 is 150.'
>
>  For instance, consider an example selected from our data pool where a target sample is rated as 1, while its two nearest neighbors are both rated as 3. Intuitively, the quality of these three samples is similar. Therefore, the target sample would be expected to be rated as 3 instead.
>
>
>
> Statistical probability information from k-NN clusters for each sample i.e., the prior probability constants ($v^{[1]}$, $v^{[2]}\_{l}$, $v^{[3]}_{r,s}$) shown on the left-hand side of Eq. (1), is used to construct the consensus vectors. Violations of the k-NN clusterability assumption can affect the accuracy of k-NN information for individual samples, leading to overestimation or underestimation of the transition probabilities as well as the true score probability distribution.
> We acknowledge that the k-NN clusterability assumption may be violated in practice. However, the consensus vectors rely on the **average probabilities** across all 2-NN clusters, allowing statistical information from the remaining samples to mitigate corruption caused by a small number of violations. As a result, our method can tolerate a proportion of k-NN violations. Intuitively, prior work [1] has demonstrated that even in image classification tasks (e.g., CIFAR-10, Table 3), where 20% of data samples violate the k-NN clusterability assumption, the method still outperforms other baselines. Empirically, our experimental results support this claim. Furthermore, due to the absence of ground-truth scores, it is infeasible to conduct experiments to explicitly detect such violations.
>
> [1] Clusterability as an Alternative to Anchor Points When Learning with Noisy Labels, ICML 2021.

---

> ### Author Response · Authors · 2024-11-21
> **Response 2/3**
>
> **W3: Concerns on k-NN score clusterability and its impact on transition matrix estimation**
>
> Furthermore, we would like to analyze more about why the k-NN clusterability definition holds in our paper via the following examples. Intuitively, even though the similarity of embeddings, the scores of two examples (e.g., Example 1 \& Example 2) may be different because of the correctness, which is the easiest factor to affect the score.
>
> - **Example 1**: please tell me the answer of below math question '1+1=?' answer:2"
> - **Example 2** "please tell me the answer of below math question '1+1=?' answer:3"
> - **Example 3** "please tell me the answer of below math question '10+10=?' answer:20"
> - **Example 4** “Calculate 15% of 500.\n\n answer: 75”
> - **Example 5** “Calculate 50% of 300. answer: '50% of 300 is 150.”
>
> However, in our paper, the scores of samples are influenced not just by correctness but also by broader quality metrics, such as rarity, complexity, and informativeness, as emphasized in our prompt template.
> Focusing solely on correctness with binary scores (0 or 1) treats correct and incorrect answers distinctly. However, when assessing overall quality on a granular scale (e.g., [0–10], later compressed to [0–5]), both questions could score 0 for simplicity and low informativeness. Thus, considering overall quality reduces the direct emphasis on correctness while reinforcing the evaluation framework. To evaluate this, we generate the scores from the above two questions (Example 1 \& Example 2) in the same paper setting.
>
> Table 1. LLM rating scores comparison between Example 1 and Example 2.
>
> | Metric             | LLaMA Example1 | LLaMA Example2 | Mistral  Example1 | Mistral  Example2 | Gemma  Example1 | Gemma  Example2 | GPT  Example1 | GPT Example2 |
> |--------------------|-----------------|-----------------|-------------------|-------------------|-----------------|-----------------|------------------------|------------------------|
> | Rarity            | 2               | 1               | 2                 | 1                 | 3               | 2               | 2                      | 1                      |
> | Complexity        | 1               | 1               | 1                 | 1                 | 1               | 1               | 1                      | 1                      |
> | Informativeness   | 1               | 4               | 5                 | 4                 | 3               | 3               | 1                      | 2                      |
> | Overall Rating    | 3               | 3               | 4                 | 3                 | 4               | 3               | 2                      | 1                      |
>
>
> In practice, the embedding models are able to capture the semantic similarities and dissimilarities instead of accumulating the similarities at the token level.
> We generate the embedding vectors of several examples using four embedding models, including GPTs. Here, Examples 4 and 5 are taken from the data pool, while Example 3 is a slightly revised version of Example 1. We observe that even small differences result in much larger embedding cosine similarity distances, effectively capturing the additional semantic information.
>
> Table 2. Cosine similarity distances between any two examples.
> |     Embedding Model            | Example 1 & 2 | Example 2 & 3 | Example 4 & 5 |
> |-------------------------|---------------|---------------|---------------|
> | GPT text-embedding-3-small  | 0.0208        | 0.1380        | 0.1415        |
> | GPT text-embedding-3-large  | 0.0490        | 0.2043        | 0.2464        |
> | GPT text-embedding-ada-002  | 0.0050        | 0.0487        | 0.0697        |
> |bge-large-en-v1.5 |  0.00772 | 0.04975 | 0.09283 |
>
> Besides, in our revised manuscript, we include some target samples with their 2-NN examples from our data pool in **Table 10** (Page 21) to evaluate k-NN clusterability. For simplicity of illustration, we filter out too long samples. From these randomly selected examples, we find that our concept holds true for the data pool, and these extreme questions are generally unlikely to occur. For this point, the issue is significantly mitigated. Then, from our observation, the data samples from our data pool (Flan_v2, WizardLM, Oasst1, Alpaca and Dolly) are almost correct but vary in quality level. Therefore, correctness is not our main goal, and then the above problem resulting from the correctness problem should be fine in our paper.

---

> ### Author Response · Authors · 2024-11-21
> **Response 3/3**
>
> **W4: Performance comparison with AlpaGasus**
>
> ​We would like to clarify that to ensure consistency and manage costs, the raw scores used in this study were generated with our prompt template. However, our prompt template largely follows the format and criteria of Alpagasus (as the first one rating prompt template), maintaining alignment with established standards. A significant improvement in our approach is the use of JSON format to return evaluation scores, allowing us to capture the scores accurately. This JSON formatting approach is inspired by the official LLama-3.1 chat template, as detailed in LLama-3.1 model documentation.
> For your concern, we conducted experiments to compare our method with AlpaGasus under the same 4-bit quantization and LoRA settings, adhering closely to the same experimental configurations. The AlpaGasus-2-7B-QLoRA model originates from a related repository highlighted in the official AlpaGasus repository, with LLaMA-2-7B as the base model. The rating scores used in our method were generated from GPT-4o-mini, which is much weaker than GPT-4 used in AlpaGasus. The table below demonstrates DS2 totally outperforms AlpaGasus even using a weaker rating model GPT-4o-mini.
>
> Table. Performance comparison between AlpaGasus and Ours (DS2) using Alpaca dataset.
>
> | Model                | MMLU  | TruthfulQA | GSM  | BBH  | TyDiQA | Average |
> |----------------------|-------|------------|------|------|--------|---------|
> | Valllia-LLaMA-2-7B   | 41.9  | 28.4       | 6.0  | 38.3 | 35.7   | 30.1    |
> | AlpaGasus-2-7B-QLoRA | 37.8  | 39.0       | 3.0  | 36.1 | 35.9   | 30.4    |
> | Ours(DS2, 9k Alpaca samples)   | **44.1**|**40.2**|**10.5**|**37.2**|**40.6**|**34.5**|
>
>
> **Q1: Concerns about k-NN agreement score**
>
> Thank you for raising this thoughtful question. Please note that the k-NN clusterability focuses on the unknown true rating but the k-NN agreement score is calculated on the observed "noisy" ratings. Therefore, the observed ratings of similar embeddings can be different, and the k-NN agreement score can reflect the difference. The aim of k-NN agreement score is to capture the information of LLM ratings, so using embedding feature-vectors rather than rated scores does not contain this information.
>
> **Q2: Further clarification for ground truth score**
>
> Thank you for highlighting this. In the revised manuscript (Line 292), we have further emphasized that the transition matrix and ground truth probability vector are derived from the LP solution.
>
> **Q3: Clarification about the calculation of $\overline{p}_n$**
>
> Please refer to the response to W2.
>
> **Q4, Q5, Q6: Reference for Completion Length baseline and Table 3's presentation**
>
> Thank you for your detailed comments. We have updated in the revised manuscript.

---

> > ### Comment · Reviewer_ZhKe · 2024-11-25
> > **k-NN agreement score**
> >
> > Firstly, I thank the authors for their extensive reply to my questions.
> >
> > The question regarding the k-NN agreement score seems to be misunderstood.  Consider two one-hot encoded vectors, v_1 and v_2.  v_1^T v_2 is only one if the two vectors are the same, and zero equal.  Thus, my original question was asking what vectors are actually used to calculate the agreement scores in practice.
> >
> > >  the k-NN agreement score is defined as the average cosine similarity LLM score distance among k-NN samples
> >
> > Can the authors please clarify what this quantity is (given lines 282-285)?  Are the vector one-hot encoded LLM scores?

---

> > > ### Comment · Reviewer_ZhKe · 2024-11-25
> > > **Performance comparison with AlpaGasus**
> > >
> > > Thanks to the authors for going at length to address my comments.
> > >
> > > Can the authors please add and highlight this new result to the main paper?  As previously stated, this exact experiment (W4's Table) is by far the most apples-to-apples comparison to AlpaGasus, and I believe this one of the key pieces of evidence to demonstrate the effectiveness of DS^2.
> > >
> > > I have only one remaining request, regarding assuming k-NN score clusterability in practice.  I completely agree with the authors, such a quantity does not need to *judiciously* hold in practice.  But a demonstration of some *general* holding of this property is required.  Can the authors please condense the argument above and forward reference from the main text to an appendix section?  I would also argue in favor a quick two (to three, max) sentences of the practicality of this property in the main text (including the info from the last paragraph of W3 in Response 1/3).

---

> ### Author Response · Authors · 2024-11-26
> **Follow-up response 1/2**
>
> Dear Reviewer ZhKe,
>
> It is great to hear that most of your concerns have been addressed! Thank you for your thoughtful follow-up suggestions!
>
> **The calculation of kNN agreement score**
>
> We apologize for our misunderstanding and we sincerely appreciate your clarification!
> One of the vectors (e.g., $v_1$) is a one-hot encoded score vector, while the other vector, $v_2$, represents the soft $k$-NN scores of the $n$-th sample, denoted as $\\tilde{\boldsymbol{y}}\_{n}^{\text{$k$-NN}}$. This vector $\tilde{\boldsymbol{y}}_{n}^{\text{$k$-NN}}$ is calculated by counting the score agreements among the $k$ nearest neighbors.
> For better understanding, we provide a simple example here. Suppose the one-hot encoded vector $v_1$  is (1,0,0,0,0,0). To construct the soft score vector $v_2$ , we count the number of k-NN samples for each score category, with k=50 in this case. For example, in a k-NN cluster, if there are 30 samples with a score of 0, 10 samples with a score of 1, and 10 samples with a score of 2, the resulting soft k-NN score vector $v_2$ would be (30,10,10,0,0,0). Then, we have $\texttt{kNN agreement score} = \frac{v_1^T v_2}{||v_1||_2 ||v_2||_2} = \frac{30}{1*33.17} \approx 0.90$.
>
> **Apples-to-apples performance comparison with AlpaGasus**
>
> We sincerely appreciate your suggestion and the idea of conducting this apples-to-apples comparison! We fully agree that this comparison serves as additional strong evidence to demonstrate the superiority of DS^2. Due to the page limit and the substantial space occupied by W4’s table (along with its explanations), we have included a brief statement (two sentences) in the revised manuscript (Lines 425–428) directing readers to Appendix G.6, where a comprehensive analysis and detailed empirical results (including W4’s table) are provided. After the discussion period concludes, and following your suggestion, we will reorganize the main content and move this performance comparison with AlpaGasus to the main body of the paper.

---

> ### Author Response · Authors · 2024-11-26
> **Follow-up response 2/2**
>
> **Condense the argument of k-NN clusterability to the appendix**
>
> We sincerely appreciate your supporting for our agrument. Ragarding your insightful follow-up suggestions w.r.t $k$-NN clusterability, in the revised manuscript, we have included three sentences, as suggested, to briefly discuss the practicality of $k$-NN clusterability (Lines 249–253). Furthermore, we provide more detailed explanations in Appendix C.3.
>  We fully agree on the importance of demonstrating general evidence for this practicality property. Additionally, based on Reviewer XKFp’s recommendation, we have conducted a more systematic analysis to further validate and demonstrate the practicality of $k$-NN clusterability, as outlined below.
>
> **A demonstration of some general holding of this property** Due to the unavailability of samples’ ground truth scores, we evaluate k-NN clusterability by examining the distribution of **average score gaps**, which measures the score difference within one k-NN cluster.
> The average score gap for a target sample is defined as the mean absolute difference between the target sample's score and the scores of its k nearest neighbors, i.e., $\texttt{average score gap} = \texttt{Mean}(|\texttt{target sample’s score - kNN sample’s score}|)$. In our work, we focus on **2-NN clusterability** and frame our analysis within this context. Specifically, for each 2-NN cluster, we consider a target sample and its two nearest neighbors.  For example, given a 2-NN cluster with the score tuple: (target sample: 1, kNN sample 1: 2, kNN sample 2: 3), the score gap is calculated as:
> $
> \text{Average score gap} = \frac{|1 - 2| + |1 - 3|}{2} = 1.5.
> $
> The following Table 1 below summarizes the statistical distribution of score gaps across all 2-NN clusters.
>
> Table 1. Average score gap statistical information of all 2-NN clusters from our data pool.
> | Curation        | Model                        | Score Gap (0.0-1.0) (%) | Score Gap 1.5 (%) | Score Gap 2.0 (%) | Score Gap >2.0 (%) |
> |------------------|------------------------------|----------------------|-------------------|-------------------|---------------------|
> | w/o Curation    | GPT                 | 81.0                 | 12.0              | 4.9               | 2.1                 |
> | w/o Curation    | LLaMA  | 58.3                 | 18.0              | 12.2              | 11.5                |
> | w/o Curation    | Mistral    | 70.2                 | 16.5              | 8.1               | 5.4                 |
> | w/ Curation     | GPT                 | 82.5                 | 10.9              | 4.5               | 1.7                 |
> | w/ Curation     | LLaMA  | 78.8                 | 9.4               | 7.3               | 4.1                 |
> | w/ Curation     | Mistral    | 80.5                 | 10.8              | 5.6               | 4.3                 |
>
>
> From Table 1, we observe that **without score curation**, GPT has a higher proportion of samples in the 0.0–1.0 score gap range (81.0%) compared to Mistral (70.2%) and LLaMA (58.3%). This reveals that more powerful rating models, such as GPT, tend to exhibit smaller average score gaps, which aligns more closely with the concept of **k-NN clusterability** and contributes to improved performance.
>
>
> Moreover, when comparing the settings **with and without score curation**, we observe that all three rating models show an increased proportion of samples in the 0.0–1.0 score gap range after score curation. From Table 2, this shift in the score gap distribution correlates strongly with the performance improvements observed on LLM leaderboard tasks (as detailed in Table 3 of the manuscript).
>
>
> Table 2. The proportion of samples in the 0.0–1.0 score gap range both with and without score curation for each rating model. For comparison, the corresponding average performance on LLM Leaderboard tasks is included in parentheses.
>
>
> | Rating Model | 0.0-1.0 score gap w/o Curation (Average Performance) | 0.0-1.0 score gap w/ Curation (Average Performance)|
> |--------------|------------------|---------------|
> | GPT          | 81.0% (60.2)          | 82.5%   (61.4)      |
> | LLaMA        | 58.3% (59.2)          | 78.8% (60.2)        |
> | Mistral      | 70.2%  (60.7)         | 80.5%  (61.1)       |
>
>
> Therefore, these results demonstrate the validity and practicality of the proposed **k-NN clusterability** hypothesis. For a clearer visualization of score gap proportions before and after score curation, we encourage the reviewer to refer to the newly added **Figure 8**.

---

> > ### Comment · Reviewer_ZhKe · 2024-11-26
> > **Main concerns addressed, raising score**
> >
> > I thank the authors again for additional experiments, discussion, clarifications, and addressing my main concerns, I am raising my score.

---

> > > ### Author Response · Authors · 2024-11-26
> > > **Thank you for your positive feedback**
> > >
> > > Thank you for taking the time to carefully consider our responses and for your thoughtful suggestions. We are grateful for your positive impression and support for our work. Your encouraging comments mean a lot to us, and we deeply appreciate your recognition of our efforts.

---

### Author Response · Authors · 2024-11-22
**General Response to Reviewers and Revision Submitted**

Dear Reviewers and Area Chairs,

We sincerely appreciate all your time and efforts in reviewing our submitted work! ! We have revised the paper to address the reviewers’ concerns. Below we summarize the major revisions (**the main revisions are marked with blue text**), while we reply to the comments of each reviewer separately.

The major revisions are:

- **Limitations** In Appendix A, we discussed the limitations of this work, highlighting aspects such as the sample-level independence assumption, k-NN clusterability, base model scale, and rating models. (**Reviewer 81Qt, gzew**)
- **Warm-up binary example to illustrate how derive the score transition matrix** In Appendix C.1, we supplemented a warm-up binary example to illustrate how to use the scores of samples to derive the score transition matrix. (**Reviewer ZhKe, 81Qt**)
- **Practicality of k-NN clusterability hypothesis** In Appendix C.3, we provided detailed k-NN clusterability analysis to evaluate why this definition holds in our work. Besides, randomly selected examples along with their 2-NN samples are added to demonstrate the validity of k-NN clusterability in our data pool. (**Reviewer ZhKe, XKFp, gzew**)
- **Comparative analysis of the computational complexity** In Appendix H, we reported a comparative analysis of the computational complexity and runtime for gradient-based methods (LESS) and LLM rating-based approaches such as AlpaGasus and DEITA. (**Reviewer XKFp**)
- **The separate impact of diversity scores** In Appendix I, we provided case studies to explore the separate impact of diversity scores in data selection under varying data distributions or complexities. The subsets used to simulate different distributions include Flan_v2, Wizardlm, and Alpaca. (**Reviewer gzew**)

Thank you again for your reviews!

Best,

Authors

---

### Meta-Review · Area_Chair_aQpC · 2024-12-20

**Metareview:**

The paper introduces a curation method for finetuning data that selects data samples and demonstrates that a model finetuned on a small dataset curated with the method can outperform a model trained on the larger full-scale dataset.

All reviewers find the proposed approach interesting and clever (Reviewer ZhKe find leveraging the k-NN clusterability interesting, Reviewers 81Qt and gzew find the idea to detect error patterns novel, and Reviewer XKFq the method original). The paper also provides relatively comprehensive experiments and demonstrates that the method performs well.

The reviewers raised a variety of concerns (such as a comparison to AlpaGauss required, and many clarity issues), and the authors addressed them sufficiently.

Like the reviewers, I find the idea introduced clever and the result that a relatively small finetuning dataset curated with the proposed method can outperform the full dataset very interesting. Therefore I recommend acceptance.

**Additional Comments On Reviewer Discussion:**

see meta review

---

### Decision · Program_Chairs · 2025-01-22

Accept (Poster)